# Quantifying the contribution of shipping NOx emissions to the marine nitrogen inventory – A case study for the western Baltic Sea

**Daniel Neumann**[1], **Matthias Karl**[2], **Hagen Radtke**[1], **Volker Matthias**[2], **René Friedland**[1], **and Thomas Neumann**[1]

[1]Leibniz-Institute for Baltic Sea Research Warnemünde, Seestr. 15, 18119 Rostock, Germany
[2]Institute of Coastal Research, Helmholtz-Zentrum Geesthacht, Max-Planck-Str. 1, 21502 Geesthacht, Germany

**Correspondence:** Daniel Neumann (daniel.neumann@io-warnemuende.de)

**Abstract.**

The western Baltic Sea is impacted by various anthropogenic activities and stressed by high riverine and atmospheric nutrient loads. Atmospheric deposition accounts for up to a third of the nitrogen input into the Baltic Sea and contributes to eutrophication. Amongst other emission sources, the shipping sector is a relevant contributor to atmospheric concentrations of nitrogen oxides ($NO_X$) in marine regions. Thus, it also contributes to atmospheric deposition of bioavailable oxidized nitrogen into the Baltic Sea. In this study, the contribution of shipping emissions to the nitrogen budget in the western Baltic Sea is evaluated with the coupled three-dimensional physical biogeochemical model MOM-ERGOM in order to assess the relevance of shipping emissions for eutrophication. The atmospheric input of bioavailable nitrogen impacts eutrophication differently depending on time and place of input. The shipping sector contributes up to $5\%$ to the total nitrogen concentrations in the water. The impact of shipping-related nitrogen is highest in the off-shore regions distant from the coast in early summer but its contribution is considerably reduced during blooms of cyanobacteria in late summer because the cyanobacteria fix molecular nitrogen. Although absolute shipping-related total nitrogen concentrations are high in some coastal regions, the relative contribution of the shipping sector is low in the vicinity to the coast because of high riverine nutrient loads.

## 1 Introduction

The ecosystem of the Baltic Sea is exposed to anthropogenic pressures (Andersen et al., 2015; Korpinen et al., 2012; Svendsen et al., 2015). One major pressure is the high input of nutrients, i.e. bioavailable nitrogen and phosphorus compounds, leading to eutrophication (Svendsen et al., 2015). The eutrophication status has improved over the past decades (Andersen et al., 2017; Svendsen et al., 2015; Gustafsson et al., 2012). However, a Good Environmental Status (GES) has not been restored yet (e.g., HELCOM, 2009). Therefore, the descriptor 5 of the Marine Strategy Framework Directive (MSFD; EU-2008/56/EC, 2008) and the Baltic Sea Action Plan (BSAP) still focus on eutrophication (HELCOM, 2007).

Riverine nutrient loads have been evaluated in detail in the past decades (Sutton et al., 2011; Nausch et al., 2017; Stålnacke et al., 1999; HELCOM, 2013a; Svendsen et al., 2015). They approximately account for $2/3$ to $3/4$ of the bioavailable nitrogen input (HELCOM, 2013a, b). In addition, atmospheric deposition accounts for $1/4$ to $1/3$ of the total loads. Therefore, atmospheric nitrogen deposition is not negligible in the context of eutrophication (Simpson, 2011; HELCOM, 2005; Svendsen et al., 2015).

Atmospheric nitrogen deposition is higher above land than above water because a higher surface roughness leads to a higher dry deposition velocity (Seinfeld and Pandis, 2016, Chap. 19). Nevertheless, coastal waters are considerably impacted by atmospheric nitrogen deposition: the largest atmospheric emission sources of oxidized and reduced nitrogen compounds are located on land (CEIP, 2018) and coastal waters are closer to these sources than open ocean waters. Additionally, some gaseous nitrogen compounds condense on coarse sea salt particles, which have a short atmospheric residence time, and, hence, deposit faster into the ocean (Paerl et al., 2002; Neumann et al., 2016). The western Baltic Sea is

a region where high amounts of bioavailable nitrogen compounds are anthropogenically emitted (HELCOM, 2013b). Therefore, relatively high impacts by atmospheric deposition can be expected in this region compared to other parts of the Baltic Sea, which is why we selected it as our area of interest.

The shipping sector is an important contributor to atmospheric nitrogen oxide ($NO_X$) air pollution in Europe and also in the Baltic Sea Region (Jonson et al., 2015; Aksoyoglu et al., 2016). Thus, it considerably contributes to nitrogen deposition – particularly at the open sea. Tsyro and Berge (1998) found that the shipping sector contributed $5\%$ to $10\%$ to the $NO_X$ deposition in the Baltic Sea in 1990. The shipping sector contributed approximately $6\%$ to the total nitrogen deposition in 2000 (HELCOM, 2005) and approximately $14\%$ to the oxidized nitrogen deposition in 2005 (Bartnicki and Fagerli, 2008). In 2010, approximately $13,500\ t/a$ and $9,500\ t/a$ of the nitrogen deposition into the Baltic Sea originated from Baltic Sea and North Sea shipping, respectively. The total atmospheric nitrogen deposition accounted for $218,600\ t/a$ and the waterborne nitrogen input for $758,300\ t/a$ (HELCOM, 2013b). A specific target for a reduction of the annual nitrogen deposition from shipping was set to $5,735\ t/a$ within the latest revision of the HELCOM Baltic Sea Action Plan (HELCOM, 2013c).

The North Sea and Baltic Sea will be declared as Nitrogen oxides Emission Control Areas (NECAs) according to the *International Convention for the Prevention of Pollution from Ships* Annex VI from 2021 onwards. That means that oceangoing ships built after 2021 have to comply with "Tier III emission thresholds" when they enter the North Sea and Baltic Sea regions. These emission thresholds force emission reductions of nitrogen oxides ($NO_X$) by $75\%$ to $80\%$ compared to the currently valid Tier I and Tier II thresholds. Hence, $NO_X$ emissions of individual ships are expected to decline from 2021 onwards. However, shipping traffic is also expected to increase in the Baltic Sea in the next decades and cargo vessels have a life time of approximately 25 to 30 years (e.g., Buhaug et al., 2009; Matthias et al., 2016; Karl et al., 2019a; Smith et al., 2014; Danish EPA, 2012). Therefore, the expected reduction in overall shipping $NO_X$ emissions is rather low in the next decade (e.g., Geels et al., 2012; Jonson et al., 2015; Hammingh et al., 2012).

Commonly, studies on atmospheric nitrogen deposition focus only on the input of bioavailable nitrogen but not on its processing in the Baltic Sea (EMEP, 2017; Bartnicki and Fagerli, 2008; Tsyro and Berge, 1998; HELCOM, 2005; Stipa et al., 2007; Bartnicki et al., 2011; Hongisto, 2014). However, the impact of one input source sector, i.e. shipping, on the marine biogeochemistry does not only depend on its annual input but also on the residence times of nutrients in the system and on the location of their deposition sites. These residence times are governed by the location and time of the nutrient release as well as by the availability of other nutrients. Hence, the amount of shipping-related nitrogen deposition relative to other nitrogen inputs is not necessarily linearly related to its impact.

This led to our central research question: **How high is the contribution of shipping-related nitrogen deposition to concentrations of total nitrogen (TN) and of individual nitrogen fractions in the western Baltic Sea?** We approached this question by the means of a modeling study.

A coupled marine physical biogeochemical model was applied to simulate the western Baltic Sea combined with a tagging method previously applied to riverine inflow (Radtke et al., 2012) and salt water inflow events (Neumann et al., 2017). Raudsepp et al. (2013) performed a similar study focusing on the impact of shipping-related nitrogen deposition on nitrogen fixation by cyanobacteria in the Gulf of Finland. Raudsepp et al. (2019) assessed shipping-related nutrient inputs – nitrogen and phosphorus from atmospheric deposition and direct discharge – into the Baltic Sea focussing rather on the Gotland Basin. The authors traced the shipping contribution by calculating the difference between two simulations with and without shipping-related nitrogen deposition but did no tagging. Tagging of atmospheric nitrogen deposition has been done for the North Sea and the English Channel in several studies (e.g., Große et al., 2017; Los et al., 2014; Troost et al., 2013; Ménesguen et al., 2018; Dulière et al., 2017). The method has also been used to tag nitrogen compounds in atmospheric chemistry transport model simulations (e.g., Brandt et al., 2011; Geels et al., 2012; Wu et al., 2011).

## 2    Materials and Methods

The marine biogeochemical modeling was done with MOM-ERGOM (Modular Ocean Model – Ecological ReGional Ocean Model). The atmospheric nitrogen deposition was calculated by the Community Multiscale Air Quality (CMAQ) modeling system, which is an atmospheric chemistry transport model. The model systems were not coupled on-line. First, simulations were performed with CMAQ. Then, simulations were performed with MOM-ERGOM using nitrogen deposition from CMAQ as forcing. Both model simulations were forced by meteorological data of the coastDat2 and coastDat3 datasets calculated by COSMO-CLM (Consortium for Small-scale Modeling – Climate Mode). Nitrogen from shipping-related atmospheric deposition was tagged in ERGOM and traced through the biogeochemical system. This procedure allowed identifying the shipping contribution to different nitrogen fractions. Shipping-related nitrogen deposition was available from the CMAQ simulations.

The MOM-ERGOM simulations with tagging of shipping-related nitrogen deposition were performed from 2006 to 2012. The model was previously spun-up for several decades without tagging. The nitrogen deposition data were only available for the year 2012. Therefore, all seven simulated years were forced by the same nitrogen deposition data. The years 2006 to 2011 were only used for the model validation

and considered as tagging spin-up. The year 2012 is used for the evaluation of the contribution of shipping-related nitrogen deposition.

## 2.1 Atmospheric Modeling

The meteorological forcing data for the MOM-ERGOM simulations were taken from the coastDat2 dataset and were calculated by COSMO-CLM (Weisse et al., 2015; Geyer, 2014; Geyer et al., 2015; HZG, 2017) using version 4.8-clm-11 (Rockel et al., 2008; Geyer and Rockel, 2013), regular lon-lat grid of $0.22° \times 0.22°$ horizontal resolution with a rotated pole at $170.0°W$ and $35.0°N$, spectral nudging applied to assimilate large-scale wind data.

The atmospheric biogeochemical forcing data for the MOM-ERGOM simulations was calculated by CMAQ. The CMAQ model is maintained and provided by the U.S. Environmental Protection Agency. For this study, we used CMAQ version 5.0.1 (Nolte et al., 2015; Foley et al., 2010; Appel et al., 2017) with the cb05tucl gas phase chemistry mechanism (Sarwar et al., 2007; Whitten et al., 2010; Yarwood et al., 2005) and aero5 aerosol chemistry, which is based on ISORROPIA v1.7 (Fountoukis and Nenes, 2007; Sarwar et al., 2011). Atmospheric particles are represented by a three-moment scheme containing three size modes (Binkowski and Roselle, 2003). The dry deposition parameterization for particulate matter is an updated version of Binkowski and Shankar (1995), which is based on Slinn and Slinn (1980) and Pleim et al. (1984). The parameterization considers gravitational settling, aerodynamic resistance above the canopy, and surface resistance. The three modes and the three moments are deposited individually. Land based emissions were aggregated with SMOKE for Europe (Sparse Matrix Operator Kernel Emissions; Bieser et al., 2011). Marine shipping emissions were calculated with the STEAM model (Ship Traffic Emission Assessment Model; Jalkanen et al., 2012) based on data of the automatic identification system (AIS). Via AIS modern ships broadcast their location, direction of travel, speed, IMO number (IMO: International Maritime Organization), and further information. Ships are considered to emit $NO_X$ but no reduced nitrogen. Sea salt emissions were calculated online (Gong, 2003; Kelly et al., 2010) without surf zone emissions (Neumann et al., 2016).

The CMAQ simulations were performed on two one-way nested model domains with increasing horizontal grid resolution (Fig. 1) and 30 vertical z-layers each. The outer model domain ($64 \times 64$ km$^2$ grid resolution) covered Europe and northern Africa. The lateral boundary conditions were taken from FMI APTA global reanalysis (Sofiev et al., 2018). The first nested model domain ($16 \times 16$ km$^2$ grid resolution) covered the North Sea and Baltic Sea regions. The latter data were used as atmospheric input data for the biogeochemical modeling experiments. The following CMAQ system variables were summed to obtain oxidized and reduced nitrogen deposition:

– Oxidized nitrogen: NO, $NO_2$, $HNO_3$, $N_2O_5$, $NO_3^-$, $NO_3$, HONO, PAN (peroxyacetyl nitrate), PNA (peroxynitric acid; only wet deposition)

– Reduced nitrogen: $NH_3$, $NH_4^+$

St-Laurent et al. (2017) considered the same CMAQ variables to calculate nitrogen deposition into the ocean but additionally estimated dissolved organic nitrogen (DON) deposition according to Zhang et al. (2012). Detailed DON deposition measurements were not available for the region of interest. Therefore, atmospheric deposition of DON was not considered in this study. Figure 2 shows the resulting annual mean nitrogen deposition in the western Baltic Sea region and the contribution from the shipping sector.

Meteorological input data for the CMAQ simulations were modeled with COSMO-CLM version 5.00_clm8 with spectral nudging (Rockel et al., 2008) on a rotated grid of $0.11°$ spatial resolution. This data set is available as coastDat3 atmosphere dataset of the Helmholtz-Zentrum Geesthacht (http://www.coastdat.de/; HZG, 2017).

Karl et al. (2019a, b) describe the model setup in more detail and present a validation for the simulation results with respect to the atmospheric deposition. The wet deposition of oxidized and reduced nitrogen was systematically underestimated at Baltic Sea stations. The reported underestimation is consistent with results of Vivanco et al. (2017). Nitrogen deposition of CMAQ simulations with very similar forcing data in the same region but in different years was evaluated. The reason for the underestimation could not be fully resolved in Karl et al. (2019a). It is assumed either that $NO_X$ to $HNO_3$ conversion is too slow – possibly because of too low ammonia background concentrations – or that the wet removal of $NH_4^+$ and $NO_3^-$ is too low. Modeled atmospheric concentrations of $NO_X$ did properly reproduce measurements at EMEP stations.

The nitrogen deposition data set was bilinearly interpolated onto the MOM-ERGOM model grid resolution and supplied as daily mean values.

## 2.2 Marine Modeling

The ocean physics were simulated with the Modular Ocean Model (MOM) version 5.1 (Griffies, 2004). The whole Baltic Sea was modeled with a horizontal resolution of 3 n.m. $\times$ 3 n.m. and 134 vertical layers. Open boundary conditions were provided as climatological data in the Skagerrak, the connection to the North Sea. A dynamic ice model simulates ice cover (fraction of grid cell area), thickness and extent. MOM has been used for several studies of the Baltic Sea and has been extensively validated (e.g., Neumann et al., 2015; Radtke et al., 2012; Schernewski et al., 2015).

The marine biogeochemical processes are simulated with the Ecological ReGional Ocean Model (ERGOM), which has

been developed at the Leibniz Institute for Baltic Sea Research Warnemünde and is still under active development (Neumann, 2000; Neumann et al., 2002; Kuznetsov and Neumann, 2013; Radtke et al., 2013; Neumann et al., 2015). It was coupled to MOM and shared the same model domain. The nitrogen deposition data were supplied in daily resolution. Riverine nutrient loads were taken from the *Updated Fifth HELCOM Baltic Sea Pollution Load Compilation* (HELCOM, 2015).

In the used ERGOM version, the biogeochemical system is represented by 31 state variables ("tracers"), of which 26 are in the water column and 5 in the surface sediment. Inorganic nutrients – i.e. nitrate ($NO_3^-$), ammonium ($NH_4^+$), and phosphate ($PO_4^{3-}$) – enter the system via river input, atmospheric deposition, and remineralization of organic matter. They are consumed by phytoplankton that are represented by large phytoplankton, small phytoplankton, and cyanobacteria. Large phytoplankton start growing at lower temperatures than small phytoplankton but process nutrients less efficiently meaning that they run into nutrient limitation more quickly than small phytoplankton. The growth of cyanobacteria depend only on $PO_4^{3-}$ and molecular nitrogen ($N_2$), which they fix to cover their nitrogen demand. Phytoplankton, including cyanobacteria, are grazed by zooplankton. Plankton respire and die. Dead plankton become detritus that sinks to the sediment. The sediment is represented by a one-layer sediment including several relevant sediment processes such as phosphate release under anoxic conditions or denitrification. Nutrients may be retained in the sediment, deeply buried, or resuspended. All state variables, processes, and constants are listed in a detailed model documentation in the Supplement.

Shipping-related atmospheric nitrogen deposition was tagged by the method described by Ménesguen et al. (2006). It has been implemented in ERGOM and used in previous studies (e.g., Neumann, 2007; Radtke et al., 2012). All state variables containing nitrogen are duplicated: one variable containing all nitrogen in the particular compound and another variable containing only the shipping-related nitrogen. The first type of state variable is denoted as "all NAME" or "$NAME_{all}$", whereas the latter type is denoted as "shipping NAME" or "$NAME_{ship}$". Process rates are calculated for the original state variables and, then, are linearly scaled according the $NAME_{ship}$-to-$NAME_{all}$ ratio of the educts (also written as $NAME_{ship}/NAME$).

Monthly mean concentrations of all state variables were written out in full spatial resolution. Basin mean concentrations were calculated from these data and, hence, are only available as monthly means. Daily mean concentrations were written out in full vertical resolution at the locations of measurement stations (see Sect. 2.5).

## 2.3    Study region

The western Baltic Sea was chosen as study region. It is bordered by land in the south, west, and northeast. Danish islands like Zealand and Funen are located in the center of this region (Fig. 3).

The land use south and west of the study region is dominated by agricultural activities, which lead to nutrient inputs into the Baltic Sea via rivers and the atmosphere. The population density is lower than along the southern North Sea but still high inducing the input of various types of pollutants – i.e. organic pollutants, heavy metals, and plastic litter. The shipping traffic volume is high because a major European shipping route leads through this region connecting harbors in the Baltic Sea to the North Sea and more distant locations. Hence, the deposition of atmospheric shipping emissions and direct discharges of ships import pollutants and nutrients into the Baltic Sea.

The seawater of the Baltic Sea is brackish with a strong gradient in the salinity starting with 20 to 25 g/kg in the Kattegat to salinities below 2 g/kg in the Bothnian Bay and in the eastern parts of the Gulf of Finland. The region of interest is characterized by strong north-south – $\approx 17$ g/kg in the north and $\approx 10$ g/kg in the Bay of Mecklenburg – and west-east gradients – $\approx 15$ g/kg in the Bay of Kiel and $\approx 8$ g/kg in the Arkona Basin. These salinity gradients affect the phytoplankton species composition: cyanobacteria grow only in regions with salinities below $\approx 11.5$ g/kg (Wasmund, 1997).

The Baltic Sea surface water is well mixed in the upper 40 m by convection and wind induced turbulence in winter (Feistel et al., 2008). No algal bloom develops as long as the water column is well mixed because algae are mixed too deep where they do not get sufficient sunlight. When the wind speeds decrease in spring, the water column becomes stratified by the development of a thermocline in 25 to 30 m depth and the temperature in the surface water rises. Hence, the beginning of the algal bloom in spring strongly correlates with calmer weather and the emergence of stratification. First, diatoms begin to bloom in the nutrient-enriched surface waters in February to May (Neumann et al., 2002). Nutrient concentrations decrease and flagellates, which are more efficient in their nutrient uptake than diatoms, start blooming in April or May and reach their peak in July. The bloom declines when one of the required nutrients is depleted in summer. The biogeochemical system is nitrogen limited in most parts of the Baltic Sea indicated by nitrogen-to-phosphorus (N:P) ratios below 16, which is the Redfield ratio (Feistel et al., 2008, Sect. 12.3, Table 12.3). Hence, excess phosphorus remains in the surface water after the diatom and flagellates blooms. The N:P ratio in riverine nutrient loads mostly is larger than 16:1 indicating phosphorus limitation (Svendsen et al., 2015). However, the areas affected by river plumes and phosphorus limitation are rather small. Cyanobacteria bloom in late summer. They fix dissolved $N_2$ and, hence, are not affected by depleted nitrate and ammonium. The algal bloom

period ends in autumn when the stratification is broken up by autumn storms.

## 2.4 The year 2012

Only one year (2012) was considered to be analyzed with this study.

In 2012 there were no exceptionally strong Baltic Sea inflows from the North Sea, which might have affected salinity, temperatures and physical parameters (Mohrholz, 2018a). The precipitation amount in Northern Europe in 2012 was above the long term average EMEP (2014, p. 49). Hence, the freshwater inputs were higher than in the previous years. The riverine nutrient loads were not exceptionally high compared to the long term average. Further, Savchuk (2018) assessed the nutrient dynamics from 1970 to 2016 based on measurement data and concluded that 2012 was no exceptional year with respect to DIN (dissolved inorganic nitrogen), TN (total nitrogen), DIP (dissolved inorganic phosphorus), and TP (total phosphorus) in the water.

The nitrogen wet deposition in Northern Europe in 2012 was above the average of the previous ten years due to the increased precipitation (EMEP, 2014). The nitrogen dry deposition in Northern Europe in 2012 was lower than in the previous ten years (higher wet leads to lower dry deposition) but the total nitrogen deposion (dry + wet) was still higher. The $NO_X$ emissions in Europe in 2012 were lower than in the previos ten years. While the deposition of oxidized nitrogen compounds in Southern Europe was lower compared to previous years due to lower emissions, it slightly increased in Northern Europe due to the higher wet deposition. The ammonia emissions are treated differently in *SMOKE for Europe* than in the EMEP emission model. Therefore, the information on reduced nitrogen deposition in EMEP (2014) is not applicable here. Unfortunately, the Emissions by *SMOKE for Europe* were specifically created for the year 2012 and are not fully comparable to previously by *SMOKE for Europe* create emissions of other years.

Summarizing, the year 2012 was no exceptional year with respect to central nutrient dynamics. In the evaluation of EMEP model results, the total nitrogen deposition in Northern Europe was slightly increased due to increased precipitation. However, this might not be the case in this study because nitrogen deposition in the CMAQ data was lower than in the EMEP data.

## 2.5 Validation and evaluation

The results of MOM-ERGOM simulations were validated against observational data at specific stations (see Table 1 and Figs. 3 and 4). Vertical profiles were measured at most stations. Measurements and model data of salinity, temperature, nitrate ($NO_3^-$), and phosphate ($PO_4^{3-}$) were averaged over the top $10\,m$ and over the bottom $8$ to $10\,m$ for this purpose. The measurement data were taken and merged from two sources:

– Measurement database of the Leibniz Institute for Baltic Sea Research (IOWDB, https://www.io-warnemuende.de/iowdb.html)

– HELCOM oceanographic measurement database hosted by the International Council for the Exploration of the Sea (ICES, http://ocean.ices.dk/helcom/Helcom.aspx)

A statistical validation of the model results with measurement data is difficult because the number of observations is limited – far below one measurement per month at most stations. Therefore, seven years of data were summarized on monthly basis to a one-year 'climatology'. Climatological median and spread ($10\,\%$- to $90\,\%$-percentiles) of the measurement and model data time series were then visually compared.

Three stations were chosen to be presented in the results section. They represent different regimes in the considered region: two offshore stations in different basins (OMBMPM2 and BY2) and one station close to the shore (DMU547). Validation plots at three additional stations are presented in the supplement and show a similar outcome.

The atmospheric shipping contribution to the nitrogen budget was assessed on the basis of (a) the listed stations and (b) horizontal mean values per basin. Basin definitions by Omstedt et al. (2000) were used for this study (Fig. 4). The definitions of the basins are based on the bathymetry: e.g. the Belt Sea and the Arkona Basin are separated by the Darss Sill, which is located a bit northward to station OMBMPM. The Kattegat is not considered because the concentrations of tagged compounds might be impacted by the model's open boundary in the Skagerrak.

**Table 1.** List of stations used for validation. The first three stations (DMU547, OMBMPM2, and BY2) are considered in the manuscript, whereas the last three stations are presented in the Supplement. See also Figs. 3 and 4 for maps containing the station locations.

| Station Name | Lon [°E] | Lat [°N] |
| --- | --- | --- |
| DMU547 | 10.09 | 55.67 |
| OMBMPM2 | 11.55 | 54.32 |
| BY2 | 14.08 | 55.00 |
| OMBMPN1 | 11.32 | 54.55 |
| OMBMPM | 12.22 | 54.47 |
| OMBMPK8 | 12.78 | 54.72 |

## 3   Results

### 3.1   Validation

Figures 5 and 6 show climatological time series generated from model and measurement data of the years 2006 to 2012. Sea surface temperature is well reproduced by MOM at all stations but sea surface salinity is overestimated at OMBMPM2 and BY2. This is a known issue and has been documented previously (e.g., Neumann and Schernewski, 2008). No measurements at the sea floor were available at DMU547. At the sea floor at OMBMPM2, the modeled salinity exceeds the measurements and the amplitude of the seasonal cycle of the modeled temperature seems to be too low. This might point to issues in the vertical transport in the Bay of Mecklenburg.

Modeled sea surface nitrate and phosphate concentrations agree well with the measurements, although phosphate concentrations are slightly underestimated. The seasonal pattern of modeled concentrations is realistic at all stations at the sea surface. At the sea floor, the annual cycle of nitrate does not seem to be captured by the model at OMBMPM2. Modeled nitrate concentrations increase in spring but measurements show a decrease. Simulated salinity suggests that stratification is overestimated by the model leading to a lower impact of surface processes on deeper water layers. This also causes the damped amplitude of the annual temperature cycle. At BY2, the annual cycle of nitrate and phosphate concentrations is reproduced by MOM-ERGOM but the nitrate depletion in spring is underestimated.

### 3.2   Spatial pattern of shipping-related nitrogen

Figure 7 provides an overview on the spatial distribution of shipping nitrogen in total nitrogen ($TN_{ship}$). The $TN_{all}$ concentrations are high in the vicinity of major river estuaries – particularly close to the Oder River in the bottom right of the plotted domain and northward of Zealand – and have a strong horizontal gradient towards the open water.

The $TN_{ship}$ concentrations are very high close to the Oder River estuary and between Zealand and Lolland. They are also slightly increased in the region around the station DMU547. The spatial pattern is more homogeneous than that of the $TN_{all}$ concentrations and it reveals considerably smaller spatial gradients. Shipping routes are not visible because $NO_X$ from shipping emissions does not necessarily deposit close to their sources but might be transported over longer distances. A reason for this is the high atmospheric residence time of $NO_X$. Possible reasons for the peaks along the shoreline are discussed in Sect. 4.

The contribution of shipping-related nitrogen to TN ($TN_{ship}/TN_{all}$) does not exceed $4\%$ on annual average. It is lowest in regions close to river estuaries ($< 1\%$) and increases towards the open water.

### 3.3   Seasonal cycle of shipping-related nitrogen

The annual cycles of nitrogen compounds in the surface layer of three basins are plotted in Fig. 8.

The concentrations of dissolved inorganic nitrogen ($DIN_{all}$) in the Belt Sea and in the Arkona Basin decrease in spring, have their minimum in summer, and increase in autumn. This is an expected and typical system behavior: DIN accumulates in winter, is consumed by phytoplankton in the growth period, and is reimported into the surface layer from below by vertical mixing in autumn. Correspondingly, the concentrations of dissolved organic nitrogen and particulate organic nitrogen ($DON_{all}$ and $PON_{all}$, resp.) rise in spring and decrease in autumn. The $TN_{all}$ concentrations, which are the sum of $DIN_{all}$, $DON_{all}$, and $PON_{all}$, slightly decrease in the course of the year.

In contrast, the $DIN_{all}$ concentrations are elevated ($\approx 5$ mmol m$^{-3}$) throughout the year in the Öresund. The seasonal patterns of the $DON_{all}$ and $PON_{all}$ concentrations are the same as at the other stations. The relative contributions of shipping N to DIN, DON and PON are very small.

In the Belt Sea, the seasonal variability of the shipping contribution and its spatial variability are very low in all nitrogen fractions. The shipping contribution is between $1\%$ and $2\%$. In the Öresund, it decreases from $1.5\%$ to $2\%$ in January to about $1\%$ in July and then increases again towards the end of the year. Finally in the Arkona Basin, the shipping contribution increases from the beginning of the year until summer and then decreases. The values are in a range between $1\%$ and $4.5\%$. However, there are some places in the Arkona Basin where the shipping contribution remains below $2\%$.

Summarizing, the three considered basins represent three different regimes of shipping-related nitrogen deposition and of its contribution to the biogeochemical cycle. However, the relevance of shipping-related nitrogen differs spatially within each basin: the shipping contribution to the nitrogen fractions is much higher in the open ocean than along the coastline.

Figures 9 and 10 show monthly median and percentiles calculated from daily mean values at the three stations (two of which are in the open ocean) in the surface and bottom layer, respectively. At the sea surface, the seasonal cycles of the $DIN_{all}$, $DON_{all}$, and $PON_{all}$ concentrations are as expected. The time series of $PON_{all}$ and $TN_{all}$ concentrations shows two peaks: the first is the diatom bloom in spring and the second a cyanobacteria bloom in later summer. Cyanobacteria do not grow in the northern Belt Sea and Kattegat because the salinity is too high.

In the surface layer, the relative shipping contribution rises in all fractions and at all stations in spring, peaks in summer, and decreases again. At BY2, the $PON_{ship}/PON_{all}$ ratio decreases already after June and has a minimum in August, after which it increases again. The minimum is caused by the cyanobacteria bloom because the cyanobacteria fix non-tagged $N_2$. The overall shipping contribu-

tion at DMU547 is similarly low as in the total basin. At OMBMPM2 and BY2, the $TN_{ship}/TN_{all}$ ratio exceeds $5\%$. At BY2, the $DIN_{ship}/DIN_{all}$ ratio even exceeds $10\%$. Thus, the shipping-related nitrogen contribution in summer is much higher at individual stations in the center of the basins than on basin average – in the surface layer.

The shipping contribution to the nitrogen fractions is much lower in the bottom layer of the three stations. It remains below $2\%$ in all nitrogen fractions at DMU547 and OMBMPM2. At BY2, the contribution is higher than $2\%$ but still considerably lower than at the surface due to vertical stratification. The $PON_{ship}/PON_{all}$ ratio peaks with $\approx 6\%$ at the bottom of BY2 in summer.

A vertically resolved meridional cross section through the Arkona Basin is evaluated in the next section (Sect. 3.4) in order to assess these differences between surface and bottom layer concentrations at station BY2 in more detail.

### 3.4    Vertical distribution of shipping-related nitrogen in the Arkona Basin

In the previous sections, the spatial and temporal distribution of shipping-related nitrogen has been assessed. In this section, a cross section through the Arkona Basin is presented in order to evaluate the vertical distribution of shipping-related nitrogen.

Figure 11 shows meridional cross sections of DIN, PON, and TN concentrations through the Arkona Basin along $14.08\overline{3}^{\circ}E$.

In winter, the Arkona Basin is vertically well mixed. A horizontal gradient clearly exists with low values in the south and high values in the north. In spring, the $DIN_{ship}/DIN_{all}$ ratio increases in the central Arkona Basin at the sea surface and a vertical gradient develops. One to two month later, the $TN_{ship}/TN_{all}$ ratio also develops a vertical gradient. This time lag is reasonable because, first, a signal appears in the DIN due to external DIN input and, then, spreads to PON and DON.

The surface layer $DIN_{ship}/DIN_{all}$ ratio increases until July exceeding $10\%$ and, then, strongly decreases. The maximum of the $TN_{ship}/TN_{all}$ is at the sea surface until June 2012 when it reaches $6\%$. In the subsequent months, the maximum migrates downward and decreases. In July, the maximum is at $\approx 15$ m depth and amounts $\approx 5.5\%$. In August, it is at $\approx 20$ m and amounts $\approx 4.7\%$. The downward migration is reasonable:

Detritus with *high* shipping contribution sinks towards the seafloor as a result of the phytoplankton bloom in spring. In the open sea in early summer, production of fresh PON decreases due to nutrient limitation. PON with a high content of non-shipping nitrogen is produced in coastal regions (nutrients supplied by rivers), is horizontally mixed from the coast towards the open sea, and sinks. As a result, the maximum of the $PON_{ship}/PON_{all}$ ratio seems to migrating downward (Fig. 11, top row). If the PON concentration is much

higher than the DIN concentration, which is commonly the case in summer, the $TN_{ship}/TN_{all}$ ratio will behave similarly to the $PON_{ship}/PON_{all}$ ratio as indicated by the bottom row of Fig. 11.

### 3.5    Summary of resuls

The validation of simulations results showed a good agreement of physical and biogeochemical model data with in-situ measurements (Fig. 5).

The wet deposition of oxidized and reduced nitrogen was systematically underestimated at Baltic Sea stations. The reported underestimation is consistent with results of Vivanco et al. (2017). Nitrogen deposition of CMAQ simulations with very similar forcing data in the same region but in different years was evaluated. The reason for the underestimation could not be fully resolved in Karl et al. (2019a).

The deposition of untagged and shipping-related nitrogen was very high along the coastline. Particularly in bights and river estuaries the nitrogen deposition was considerably high. Reasons for this are discussed in the Discussion section below.

The concentration of shipping-related total nitrogen ($TN_{ship}$) was relatively homogeneously distributed horizontally (Fig. 7). A few coastal regions showed increased $TN_{ship}$ concentrations. Relatively, the contribution of shipping-related nitrogen to TN ($TN_{ship}/TN_{all}$) was highest distant from the coast due to the lack of riverine nitrogen sources in these areas.

In the surface waters of the Arkona Basin and of the Bay of Mecklenburg, the shipping contribution to all nitrogen fractions was highest in summer and lowest in winter (Fig. 8). The contribution of shipping-related nitrogen to particulate organic nitrogen ($PON_{ship}/PON_{all}$) strongly decreased in the Arkona Basin in August caused by a cyanobacteria bloom. In the bottom water, the shipping contribution was quite constant all over the year due to stable vertical stratification during the bloom period. An exception was $PON_{ship}/PON_{all}$ in summer in the bottom water of the station BY2, which is located in the center of the Arkona Basin (Fig. 10). It was caused by large amounts of sinking detritus.

In the Öresund, the annual cycle of the shipping contribution to all nitrogen fractions was inverted to the cycle in the Arkona Basin (Fig. 8). In the summer, the cycle shows a minimum in the Öresund and a maximum in the Arkona Basin. Particularly in summer, atmospheric deposition is an important nutrient source for large basins such as the Arkona Basin. The ratio between sea surface area and coastline length is lower in the Öresund than in the Arkona Basin leading to a lower relevance of atmospheric nitrogen deposition compared riverine nutrient loads and causing the inverted annual cycle. No clear annual cycle was recognizable in the Belt Sea. The Belt Sea is a quite diverse and complex region. Hence, one can expect that the shipping-contribution is not uniform all over the whole water body of the Belt Sea. Thus,

one might split the Belt Sea in several regions in future studies.

In vertical direction, the contribution of shipping-related N to DIN ($\mathrm{DIN_{ship}/DIN_{all}}$) was highest at the sea surface during the algal bloom period. The maximum of $\mathrm{TN_{ship}/TN_{all}}$ was at the sea surface until June and, afterwards, moved downwards due to large amounts of sinking detritus with shipping-related nitrogen.

## 4    Discussion

### 4.1    Discussion of the validation

The validation of simulations results showed a good agreement of physical and biogeochemical model data with in-situ measurements (Fig. 5). The seasonal cycle was well reproduced. Modeled $\mathrm{NO_3^-}$ and $\mathrm{PO_4^{3-}}$ concentrations deviated from measurements at the sea floor of station OMBMPM2 (Bay of Mecklenburg) in spring and autumn, respectively (Fig. 6).

The stations OMBMPN1 and OMBMPM (see Supplement) show similar deviations in the nutrient concentrations at the sea floor as OMBMPM2. These stations are located north-westward and north-eastward of OMBMPM2, respectively, close to the boundaries of the Bay of Mecklenburg. However, the deviations of modeled concentrations from measurements at these two stations are smaller than at OMBMPM2 indicating that mainly the Bay of Mecklenburg is affected.

MOM does not predict the vertical location of the halocline accurately in this region as we know from previous studies. This might affect the vertical transport – too weak vertical mixing – and lead to higher nutrient concentrations at the sea floor. Another reason might be that nutrients are released from the sediment into the water column in this region. The latter hypothesis cannot be tested because no sediment measurement data with high temporal resolution were available at this station.

The sea surface concentrations and their annual cycle seem to be well reproduced at all three stations – OMBMPM2, OMBMPN1, and OMBMPM. Hence, we assume that the observed deviations at the sea floor of the Bay of Mecklenburg do not negatively affect the general results of this study.

Very high PON and TN concentrations occurred at the station DMU547 in September and December. These were caused by resuspension of detritus through high current velocities at the sea floor (see plots in the Supplement for details).

### 4.2    Discussion of atmospheric nitrogen inputs

The wet deposition of oxidized and reduced nitrogen was systematically underestimated at Baltic Sea stations. The reported underestimation is consistent with results of Vivanco et al. (2017). Nitrogen deposition of CMAQ simulations with very similar forcing data in the same region but in different years was evaluated. The reason for the underestimation could not be fully resolved in Karl et al. (2019a).

The deposition of untagged and shipping-related nitrogen was very high along the coastline. Particularly in bights and river estuaries the nitrogen deposition was considerably high. This is partly of artificial origin and parly a result of specific atmospheric processes as we will describe below.

Atmospheric nitrogen deposition is higher above the land than above the ocean (Seinfeld and Pandis, 2016). Hence, there is a steep gradient in the nitrogen deposition away from the coastline. Coarser horizontal grid resolution of the CMAQ setup compared to the MOM-ERGOM setup and subsequent interpolation of nitrogen deposition data over the land-sea interface cause a smoothing of nitrogen deposition in this region leading to artificillay enhanced deposition into the coastal waters.

The second reason, which is non-artificial, is probably the interaction in the atmosphere between nitrogen oxides ($\mathrm{NO_X}$) from shipping, ammonia ($\mathrm{NH_3}$) from agricultural activities and animal livestock, and sea salt particles emitted from the sea surface. Although this topic is not in the focus of this study, we described some details in the subsequent paragraphs.

The $\mathrm{NO_X}$ reacts to $\mathrm{HNO_3}$. $\mathrm{HNO_3}$ condenses on wet particles and reduces the pH of the particle water (Reaction R1). $\mathrm{NH_3}$ condenses on wet particles and increases the pH of the particles' water (Reaction R2). Both processes are equilibrium processes. When both processes take place at the same time, then the pH is kept on a roughly constant level shifting the equilibirum towards the right side Reaction R3.

$$HNO_{3\ (g)} \rightleftharpoons NO_{3\ (aq)}^- + HO_{3\ (aq)}^+ \tag{R1}$$

$$NH_{3(g)} \rightleftharpoons NH_{4\ (aq)}^+ + OH_{(aq)}^- \tag{R2}$$

$$HNO_{3(g)} + NH_{3(g)} \rightleftharpoons NO_{3\ (aq)}^- + NH_{4\ (aq)}^+ \tag{R3}$$

Additionally, sodium chloride ($\mathrm{NaCl}$; $\mathrm{Na^+Cl^-}$) favors the condensation and deprotonization of atmospheric acids, such as $\mathrm{HNO_3}$ (Reaction R4). The condensation of $\mathrm{HNO_3}$ reduces the pH of the particles' water. Hydrochloric acid $\mathrm{HCl}$ is a weaker acid than $\mathrm{HNO_3}$. Hence, $\mathrm{Cl^-}$ has a higher probability to accept a proton (and to evaporate subsequently) than $\mathrm{NO_3^-}$.

$$HNO_{3(g)} + Na_{(aq)}^+ + Cl_{(aq)}^- \rightleftharpoons NO_{3\ (aq)}^- + Na_{(aq)}^+ + HCl_{(g)} \tag{R4}$$

Sea salt emissions considerably contribute to the atmospheric particle load in the vicinity to the shoreline and favor the formation of particulate nitrogen compounds. Sea salt

particles are relatively large and, hence, have a short atmospheric residence time meaning they are quickly deposited. Therefore, shipping-related nitrogen deposition is expected to be enhanced in some coastal regions through the interaction of shipping-related $NO_X$ and sea salt particles.

### 4.3  Discussion of the shipping contribution

The concentration of shipping-related total nitrogen ($TN_{ship}$) was relatively homogeneously distributed horizontally (Fig. 7). A few coastal regions showed increased $TN_{ship}$ concentrations. Relatively, the contribution of shipping-related nitrogen to TN ($TN_{ship}/TN_{all}$) was highest distant from the coast. The pattern agrees with regions of high shipping activity. This is a coincidence no interdependence. This agreement results not from shipping activity but rather from the lack of riverine nitrogen sources in offshore regions.

In the surface waters of the Arkona Basin and of the Bay of Mecklenburg, the shipping contribution to all nitrogen fractions was highest in summer and lowest in winter (Fig. 8). The contribution of shipping-related nitrogen to particulate organic nitrogen ($PON_{ship}/PON_{all}$) strongly decreased in the Arkona Basin in August caused by a cyanobacteria bloom. In the bottom water, the shipping contribution was quite constant all over the year due to stable vertical stratification during the bloom period. An exception was $PON_{ship}/PON_{all}$ in summer in the bottom water of the station BY2, which is located in the center of the Arkona Basin (Fig. 10). This is discussed further below.

In the Öresund, the annual cycle of the shipping contribution to all nitrogen fractions was inverted to the cycle in the Arkona Basin (Fig. 8). In the summer, the cycle shows a minimum in the Öresund and a maximum in the Arkona Basin. Particularly in summer, atmospheric deposition is an important nutrient source for large basins such as the Arkona Basin. The fraction between sea surface area and coastline length is lower in the Öresund than in the Arkona Basin. Moreover, the Öresund is considerably impacted by nutrient loads from the Swedish mainland and from Zealand (Mølleåen River). This leads to a lower relevance of atmospheric nitrogen deposition compared riverine nutrient loads and causing the inverted annual cycle.

No clear annual cycle was recognizable in the Belt Sea. The Belt Sea is a quite diverse and complex region. Hence, one can expect that the shipping-contribution is not uniform all over the whole water body of the Belt Sea. Thus, one might split the Belt Sea in several regions in future studies.

A vertically resolved meridional cross section through the Arkona Basin was analyzed. During the algal bloom period, the contribution of shipping-related N to DIN ($DIN_{ship}/DIN_{all}$) was highest at the sea surface in the center of Arkona Basin and decreased vertically downward (Fig. 11). During other times of the year, it showed a weak vertical gradient. The $DIN_{ship}/DIN_{all}$ ratio decreased from the center of the Arkona towards the coast – particularly towards south. The Oder River is located in the south contributing large amounts of riverine DIN and, hence, causing the low $DIN_{ship}/DIN_{all}$ values.

The maximum of $TN_{ship}/TN_{all}$ and $PON_{ship}/PON_{all}$ was at the sea surface until June and, afterwards, moved downwards due to large amounts of sinking detritus with shipping-related nitrogen. This caused the high $PON_{ship}/PON_{all}$ ratio at the seafloor at stationi BY2 (Fig. 10).

### 4.4  Comparison to other studies

Raudsepp et al. (2013) and Raudsepp et al. (2019) performed similar studies. Raudsepp et al. (2013) focused on the impact of shipping-related nitrogen deposition on nitrogen fixation by cyanobacteria in the Gulf of Finland. Raudsepp et al. (2019) assessed the impact of shipping-related nutrient inputs (direct discharge and deposition of atmospheric emissions; nitrogen and phosphorus) on biogeochemical system in the whole Baltic Sea with a focus on the HELCOM station BY15 in the Eastern Gotland Basin. The authors did not tag shipping-related nitrogen but performed two simulations: one with and another one without shipping nitrogen contribution and calculated the difference. Raudsepp et al. (2019) used the same atmospheric deposition dataset, a comparable ERGOM version but another ocean circulation model. Hence, also the year 2012 was assessed.

Raudsepp et al. (2019) found an increase of shipping-related nitrogen in $NO_3^-$, diatoms, and flagellates in early summer followed by a steep decline in late summer, which is caused by a cyanobacteria bloom. This result is comparable to this study's result at BY2 were the cyanobacteria bloom had a similar effect. At the other two stations in this study the physical conditions do not allow cyanobacteria blooms and, hence, do not show this result. The spatial pattern of the contribution of shipping-nitrogen to $NO_3^-$ and DIN in Raudsepp et al. (2019) and in this study, respectively, is very similar with respect to increased shipping-nitrogen in some coastal regions. This can be expected due to similar nitrogen deposition data sets.

In this study, shipping-related nitrogen was tagged in one simulation. The biogeochemical system behaved in the same way as when shipping-related nitrogen had not been tagged. In contrast, two simulations with and without shipping-related nitrogen inputs were performed in Raudsepp et al. (2019). The results were subtracted from each other in a second step and the difference was evaluated ("*difference-approach*"). The two simulations might reveal different system dynamics because the biogeochemical system is a complex non-linear system. Hence, the results of this study are not one-to-one comparable to those of Raudsepp et al. (2019).

Both approaches to assess the contribution of shipping activities to the nitrogen cycle are valid but should be used

for different research questions. The difference-approach by Raudsepp et al. (2019) is very useful when we want to assess *what would happen if* shipping emissions are reduced or totally avoided. The system behavior might change in this situation and, hence, two distinct simulations should be performed. The tagging-approach would no capture the non-linear changes in the system behavior. In contrast, the tagging-approach is reasonable when we want to assess the contribution of one or more nutrient sources to state variables *in the current situation*. We are mainly interested in the latter aspect and, hence, have choosen the tagging approach in this study.

## 5  Conclusions

Following Raudsepp et al. (2013), Neumann et al. (2018), and Raudsepp et al. (2019) this is the fourth study dealing with tracing shipping-related nitrogen inputs in the Baltic Sea biogeochemical system. This study focused on the western Baltic Sea using a state-of-the-art biogeochemical model.

The absolute contribution of the shipping sector to TN was highest along the shoreline, which was caused by the interaction of shipping-related $NO_X$ with sea salt particles and ammonia in the atmosphere and subsequent dry deposition. However, the relative contribution of the shipping sector to TN showed an inverted pattern: lowest contribution along the shoreline and increasing towards the open sea. Riverine nutrient inputs led to a relatively low relevance of atmospheric shipping-related inputs along the shoreline . Hence, offshored regions rather than coastal regions might benefit from reduced inputs of shipping-related nitrogen.

The contribution of shipping-related nitrogen to TN was below $5\%$ on large scale on annual average. Hence, measures like nitrogen emission control areas, which limit the $NO_X$ emissions of ships, are expected to have a low impact on eutrophication on large scale. However, the shipping contribution to TN exceeded $5\%$ in the centers of the Basins in summer. The shipping-related DIN was even in a range between $10\%$ and $15\%$ in the center of the Arkona Basin. Hence, the shipping sector – and atmospheric deposition in general – is an important nutriert source in offshore regions in summer.

The vertical distribution of nitrogen indicated that sinking of detritus leads to the transport of shipping-related nitrogen into sediment. An assessment of the sedimentary nitrogen composition is not reasonable in this study due to the simple sediment parameterization used. Hence, it is not clear what part of shipping-related nitrogen is buried in the sediment and what part is released back into the water column – either as bioavailable nitrogen or as $N_2$. Future studies should focus on the sediment – i.e. with a more sophisticated sediment model.

The contribution of shipping-related nitrogen to TN seems to be low taking values below $5\%$ on average. However, we do not have comparable numbers of the contribution of other atmospheric nitrogen emission source sectors, i.e. road traffic ($NO_X$), power production ($NO_X$), and livestock farming (ammonia/ammonium, $NH_3/NH_4^+$). In this context, this study is rather one case study. Future studies should target several source sectors in order to be able to put the relative contributions of individual source sectos into context.

*Code and data availability.* **Code:** The original MOM code is accessible via the MOM GitHub repository (https://mom-ocean.github.io/). The ERGOM code and a description of the model processes and constants are attached in the supplement. Additionally, ERGOM is available via the ERGOM homepage (https://ergom.net). Information on the technical aspects of coupling ERGOM to MOM are provided on request.

**Model output data:** The model output data are published at the World Data Center for Climate Data (WDCC) of the German Climate Computing Center (DKRZ, Deutsches Klimarechenzentrum): Neumann et al. (2019).

**Model input data:**

– The meteorological input data for MOM-ERGOM were taken from the coastDat2 database of the Helmholtz-Zentrum Geesthacht (https://www.coastdat.de/). The data are available at the WDCC of the DKRZ (https://doi.org/10.1594/WDCC/coastDat-2_COSMO-CLM).

– The CMAQ nitrogen deposition data are available upon request from the co-authors of the HZG. Some results of the CMAQ simulations are available via the SHEBA THREDDS server http://sheba.hzg.de/thredds/catalog/publicAll/WP2-Air/catalog.html.

**Measurement data:**

– HELCOM data are available via the ICES homepage: http://ocean.ices.dk/helcom/Helcom.aspx

– IOWDB data are available on request (https://www.io-warnemuende.de/iowdb.html). Please contact authors to get access to the database.

*Author contributions.* D.N. was responsible for overall structure and for writing the manuscript. He performed the MOM-ERGOM model simulations and did major programming and plotting tasks. H.R. implemented the tagging method and a tool for model validation. He contributed to the Materials & Methods and Results & Discussion sections. M.K. performed CMAQ air quality model simulations and evaluated meteorological forcing data and nitrogen deposition data. He contributed to the Materials & Methods and Results & Discussion sections. He further helped developing the research questions. V.M. contributed to the state of knowledge, to the Introduction section and to the development of the research question. He further provided input data for the CMAQ model simulations. R.F. provided measurement data, participated in the evaluation of the model data, and contributed to the Results & Discussion sections. T.N. supported developing the research question, contributed to Introduction, Materials & Methods, and Conclusions sections, and is the lead developer of ERGOM.

*Competing interests.*  The authors declare that they have no conflict of interest.

*Acknowledgements.*  Parts of the research published in this publication were carried out in the research projects MeRamo (funded by BMVI, FKZ 50EW1601) and SHEBA (Sustainable Shipping and Environment of the Baltic Sea region, EU BONUS Project, Call 2014-41). The BONUS SHEBA project was supported by BONUS (Art 185), funded jointly by the EU and national funding institutions. The MOM-ERGOM model simulations were performed at the cluster Konrad of the North-German Supercomputing Alliance (HLRN, project ID mvk00054) within MeRamo. The meteorological and atmospheric chemistry transport model (CTM) simulations were performed for SHEBA at the German Climate Computing Center (DKRZ) within the Project "Regionale Atmosphärenmodellierung" (Project ID 302), which is funded by the Helmholtz Association. The emissions for the CTM simulations were kindly provided by Johannes Bieser, Armin Aulinger, and Jukka-Pekka Jalkanen. The meteorological input data for the MOM-ERGOM simulations were taken from the CCLM coastDat2 data set by Beate Geyer. MOM has been developed and is maintained by the Geophysical Fluid Dynamics Laboratory (GFDL) which is part of the U.S. National Oceanographic and Atmospheric Agency (NOAA). The air quality model is developed and maintained by the U.S. Environmental Protection Agency (US EPA). We thank our colleagues conducting IOW's Baltic Monitoring and long-term data program, whose intense quality checked measurements we used for the model validation. Additional measurement data were kindly provided by the HELCOM oceanographic measurements database hosted by ICES. We thank Uwe Schulzweida, Charlie Zender, Paul Wessel, the R Core Team, and the Unidata development team (and all involved developers/contributors) for maintaining the open source software packages Climate Data Operators (cdo), the NetCDF Operators (NCO), Generic Mapping Tools (GMT), the statistical computing language R, and NetCDF, respectively.

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

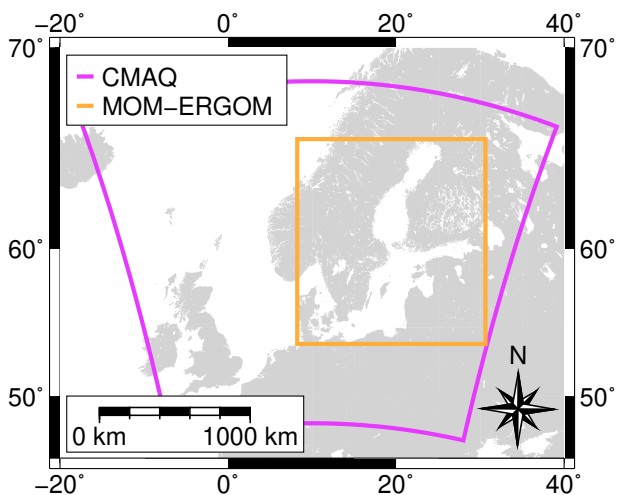

**Figure 1.** Extent of the model domains of the atmospheric chemistry transport model (CMAQ) and of the marine biogeochemical model (MOM-ERGOM).

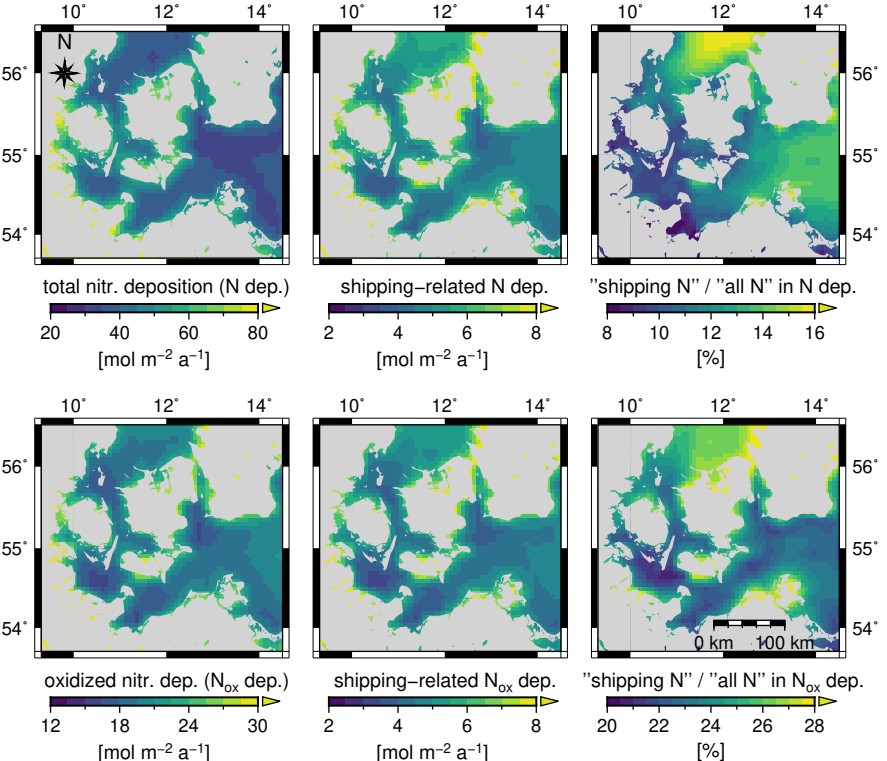

**Figure 2.** Annual mean deposition of total nitrogen (TN, top) and oxidized nitrogen ($NO_X$, bottom) calculated by the CMAQ model. The nitrogen deposition (nitrogen from all sources, left), the shipping-related nitrogen deposition (center), and the quotient between both (right) are plotted.

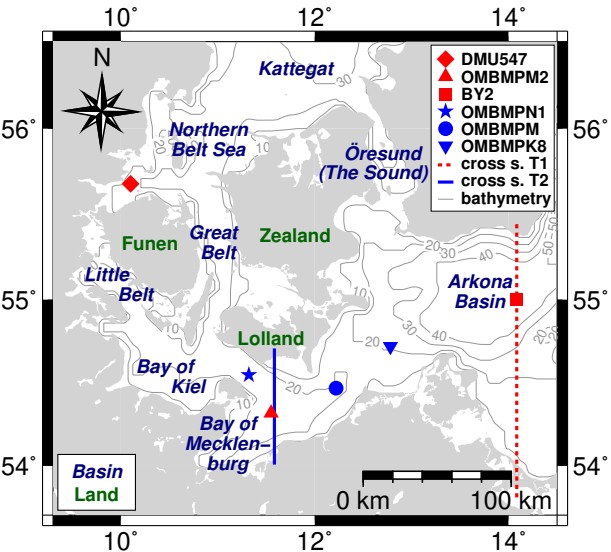

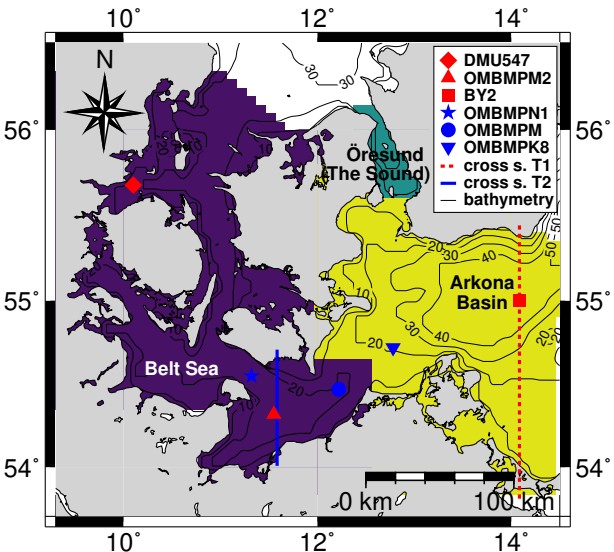

**Figure 3.** Study region, measurement stations, cross sections for evaluation, and geographic locations mentioned in this publication. Basins of the Baltic Sea and of the Kattegat are printed in navy blue and italics font. Islands and peninsulas are printed in green. Measurement stations are indicated by symbols and colors as defined in the legend top right. Cross sections for model evaluation are indicated by red dotted (T1) and blue solid lines (T2). Red stations/lines are considered in this manuscript (top three stations in the legend) and blue stations/lines in the Supplement (items four to six in the legend). The solid gray lines with numbers attached are isolines of the bathymetry. The numbers give the depth in m.

**Figure 4.** Basins in the western Baltic Sea according to Omstedt et al. (2000) drawn in the colors yellow (Belt Sea), green (Arkona Basin), and cyan (Öresund). Measurement stations are indicated by red and blue symbols. Cross sections for model evaluation are indicated by red dashed and blue solid lines. Red stations/lines are considered in this manuscript (top three stations in the legend) and blue stations/lines in the Supplement (items four to six in the legend). The dark thin lines with numbers attached are isolines of the bathymetry. The numbers give the depth in m.

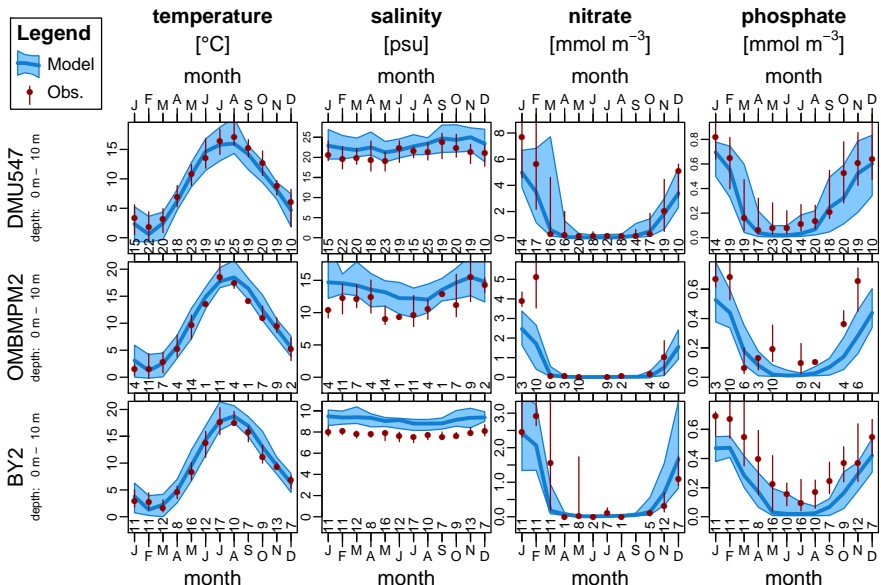

**Figure 5.** Monthly climatological medians of observational (dots) and modeling data (solid lines) at the sea surface (top 10 m) from 2006 to 2012. Each row presents the data of one station ordered from west (top) to east (bottom). The station names and depth range are given on the left. Each column presents one state variable: salinity, temperature, nitrate, and phosphate (from left to right). Vertical lines through the dots and shaded area around the solid line show the monthly variability represented by the 10% and 90% percentiles. The number of observational data points per month is given above the x-axis of each plot. A similar figure showing data at the other three stations is provided in the Supplement.

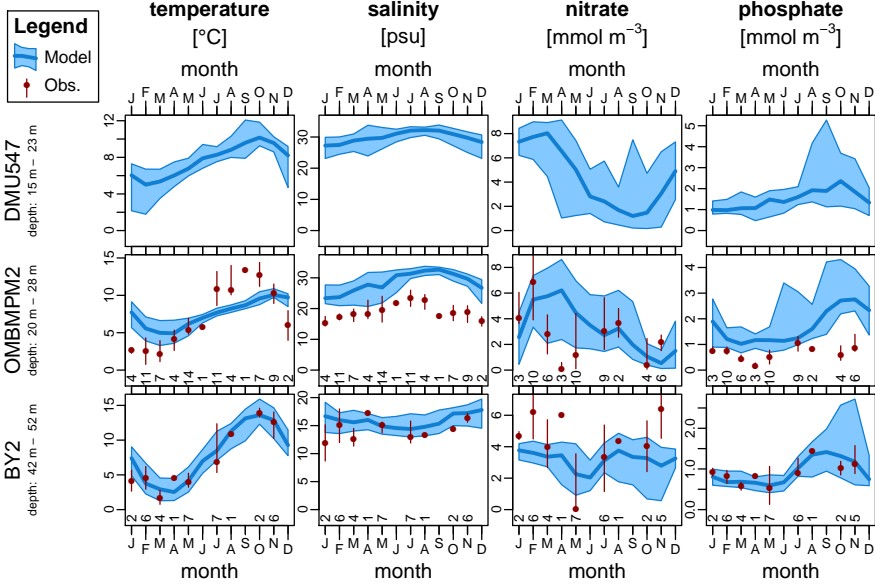

**Figure 6.** Similar to Fig. 5 but for the bottom 8 to 10 m. The exact depth range is given on the left. A similar figure showing data at the other three stations is provided in the Supplement.

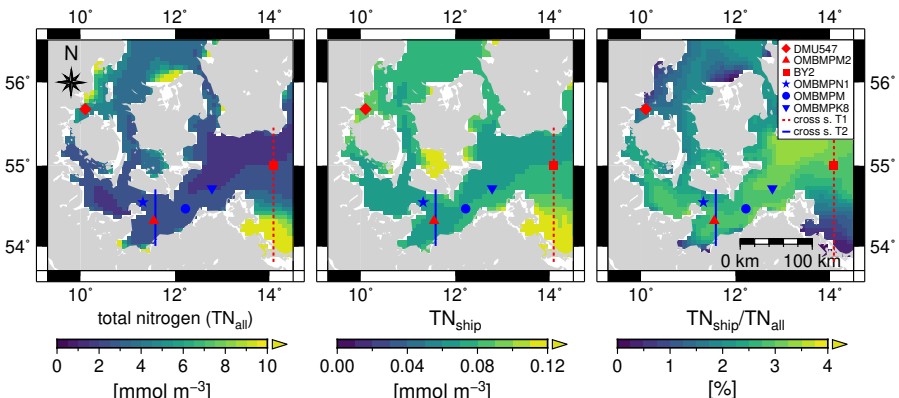

**Figure 7.** Spatial pattern of the modeled total nitrogen concentration ($TN_{all}$), the total nitrogen concentration with nitrogen from shipping-related atmospheric deposition ($TN_{ship}$), and ratio of both ($TN_{ship}/TN_{all}$). Modeled data of the year 2012 of the top 10 m are used. Measurement stations and cross sections used in other parts of the evaluation are included as symbols and lines, respectively.

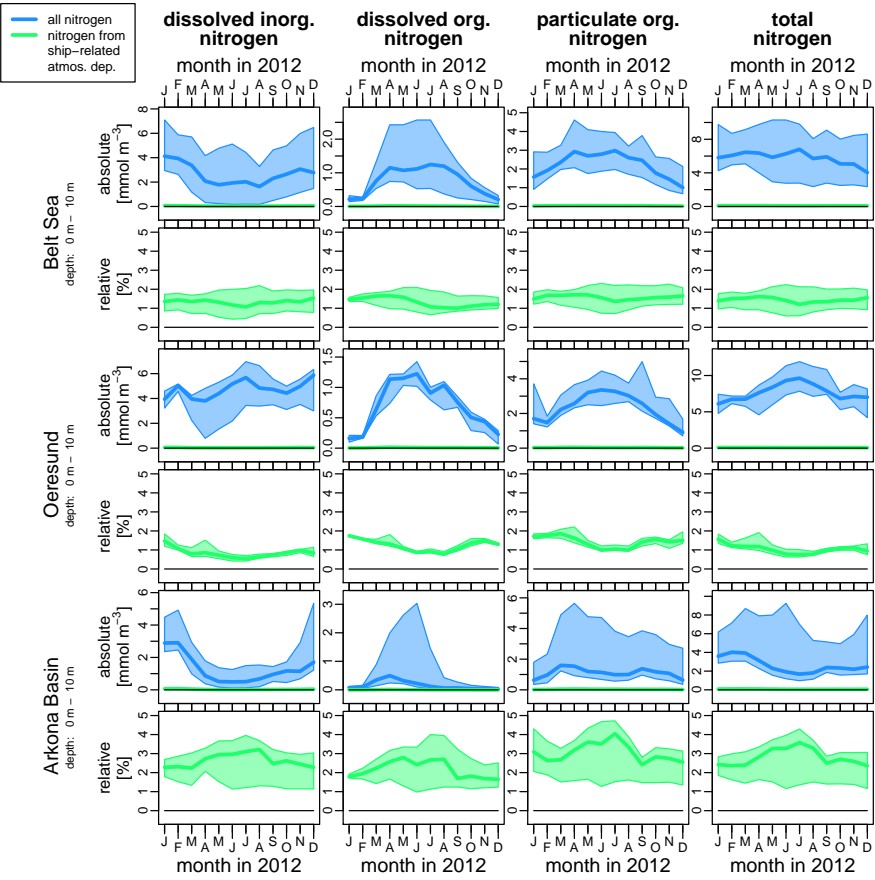

**Figure 8.** Monthly concentrations of dissolved inorganic nitrogen (DIN), dissolved organic nitrogen (DON), particulate organic nitrogen (PON), and total nitrogen (TN) with all nitrogen (blue; darker color in grayscales) and shipping-related nitrogen (green; lighter color in grayscales) in the odd rows. Ratio between shipping-related and all nitrogen of the same compounds in the even rows. Each pair of rows represents data of one of the basins Belt Sea, Öresund, and Arkona Basin (top to bottom). Horizontal median, 10 %-percentiles and 90 %-percentiles are plotted. The thick lines are the medians and the shaded area covers the interval between the 10 %- and 90 %-percentiles. The statistics were calculated from monthly and vertically (top 10 m) averaged concentrations. For the ratios, (1st) the vertical and temporal averages, (2nd) the quotients, and (3rd) the median and percentiles were calculated.

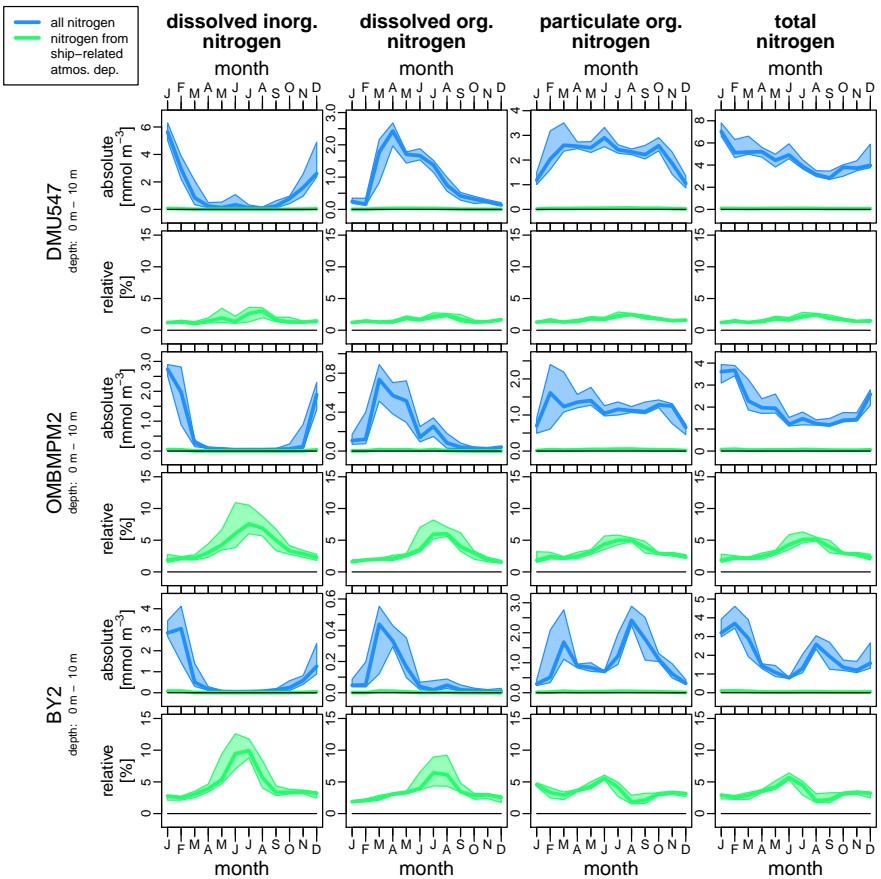

**Figure 9.** Similar to Fig. 8 but showing monthly median and percentiles calculated from daily data at specific station locations (Fig. 8 showed monthly percentiles calculated from monthly mean data and showed the variability in space). The stations are the same as in the validation. A similar figure showing data at the other three stations is provided in the Supplement.

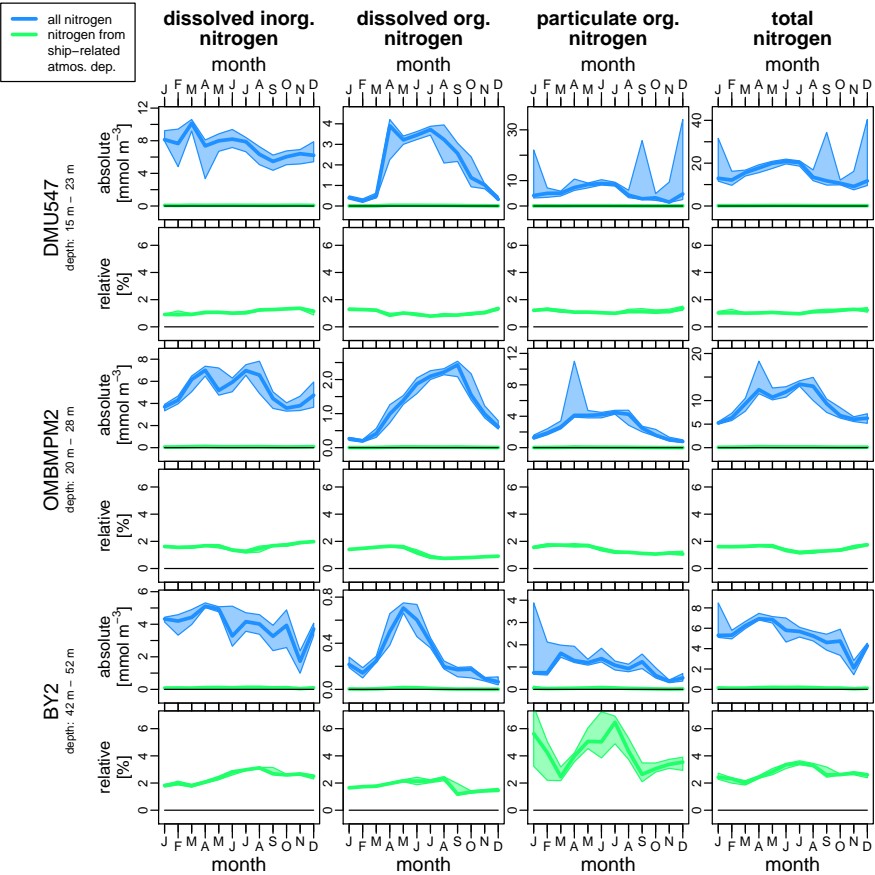

**Figure 10.** Similar to Fig. 9 but for the bottom 8 to 10 m. The exact depth range is given on the left. A similar figure showing data at the other three stations is provided in the Supplement.

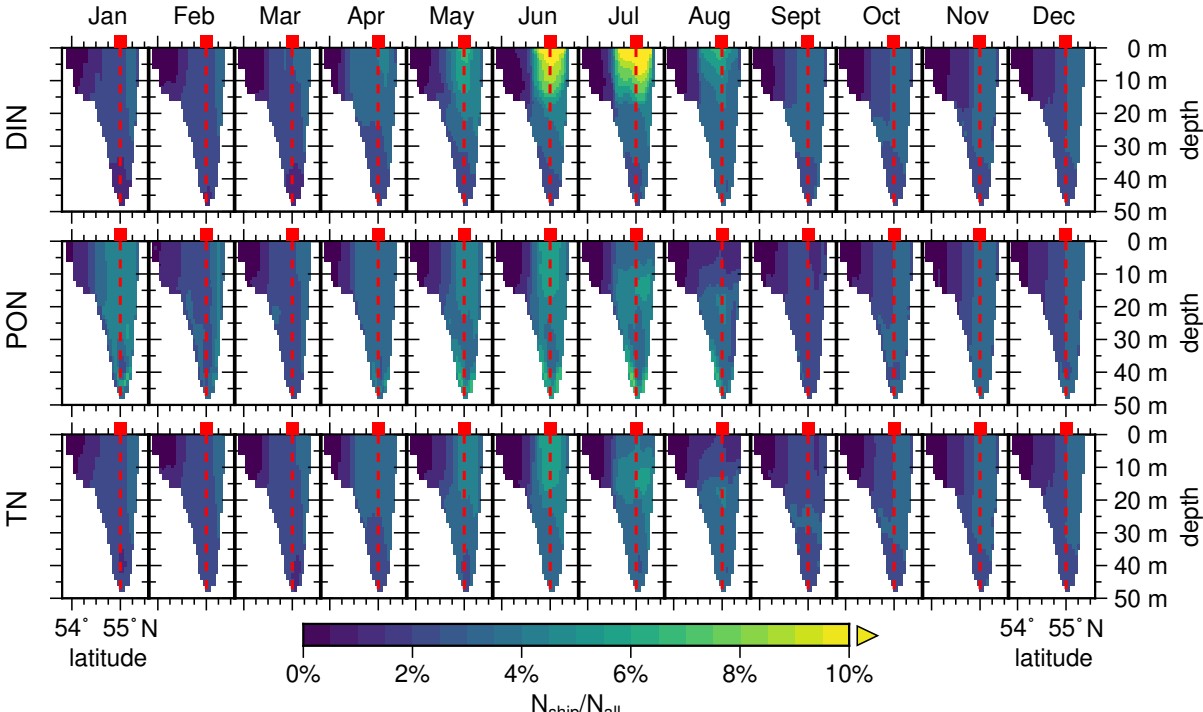

**Figure 11.** Cross section at 14.08333 °N through the Arkona Basin (red line in Fig. 3) showing the contribution of shipping nitrogen to all nitrogen in dissolved inorganic nitrogen (DIN, top) and total nitrogen (TN, bottom). Each column shows data of one month: January to December 2012 from left to right. The location of the station BY2 is indicated by a red symbol at the sea surface. The vertical red dashed line represents the measurement profile taken at this station.