# Peer review of "Quantifying the contribution of shipping NOx emissions to the marine nitrogen inventory – A case study for the western Baltic Sea"

_Ocean Science, 2019_

## Referee Comment (RC1) · Fabian Große (Referee) · 16 Jul 2019

**Review of "Contribution of shipping NOx emissions to the marine nitrogen budget of the western Baltic Sea – A case study" by Neumann et al.**

This study presents a model-based analysis of the impact of ship-borne atmospheric nitrogen (N) on the biogeochemistry in the western Baltic Sea, a region with significant shipping traffic and in reach of shipping emissions in the North Sea and Baltic Sea. The authors combine the physicalbiogeochemical MOM-ERGOM model with an element tagging technique to track ship-borne atmospheric N through the environment. Forcing fields of (ship-borne) atmospheric N deposition are provided by the CMAQ atmospheric model.

The authors provide a sound model validation and conduct a detailed analysis for the year 2012. They find that, on a regional/basin scale, the contribution of ship-borne N to near-surface total nitrogen (TN) stays well below 5%. However, its contributions to dissolved inorganic (DIN) and organic nitrogen (DON) can exceed 10% at some offshore locations not affected by riverine N inputs during the seasonal cycle. Hence, the shipping sector plays a small, yet non-negligible role for the biogeochemistry in the western Baltic Sea.

The manuscript is generally well written, concise and easy to read. I do have one major concern, which is that the manuscript lacks are thorough discussion of the results, both with respect to the limitations of the study and in the context of existing literature. Another a little bit bigger point relates to the manuscript title and the presented results. From my point of you, this study only quantifies the ship-borne contribution to the N **inventory**, but not to the **budget**, because only N variables (or sums of such) and no processes are considered. Although the latter is not absolutely necessary for this study, a TN budget for overall and ship-borne N could be a very nice and interesting contribution.

Aside from that, I have mostly minor suggestions/technical corrections. Therefore, I recommend reconsideration for publication after moderate revisions.

**General comments**

The "Summarizing Discussion" is less of a discussion but more of a summary. Results should be discussed in the context of existing literature (e.g. comparison with the approach of Raudsepp et al. (2013) mentioned in the Introduction; is the tagging approach better/preferable? Why?). It needs to be very clear what the main insights and contributions of the present study are, and how they expand on previous knowledge. In addition, the limitations of the study need to be discussed. For instance, does the year 2012 represent an average year in terms of environmental conditions, inputs from different types of nutrient sources etc. or was it an exception with respect to some factors? If the latter is the case, what implications could that have on the generality of the results. Some limitations of the model are touched in the course of the Results section but those get a bit lost and they are not sufficient for an in-depth discussion. The authors further emphasize the role of the sediment in the Conclusions. So, what influence of the relatively simple sediment parameterization can be expected with respect to the presented results? (Note that this list of potential discussion points is not meant to be exhaustive.) The authors may want to consider separating Results from Discussion more clearly.

As mentioned above, an actual TN budget for overall and ship-borne N (e.g. for the basins defined in Fig. 4) could be a very nice addition to the manuscript. Or even a budget of the overall and shipborne N fluxes between the different model state variables (similar to a model schematic with numbers for the different fluxes for overall and ship-borne N). Since the manuscript is quite concise in its current form, this would not make it overly lengthy. If it's not possible to calculate such budget, i.e. the required fluxes were not stored, the authors should consider changing the title by using "inventory" instead of "budget". The Results section is a bit of a laundry list without a clear transitions between the individual results describing why the specific (upcoming) results are shown. Including such transitions from one subsection/result to the next would help the reader to get what the key focus of each figure is. Also, as a general comment on the results and the discussion: please regularly include cross-references to figures and figure panels to allow for an easier link between text description and figures. You may want to add panel labels (a, b, c, ...) for easier in-text referencing.

I don't think all additional figures provided in the supplement are necessary, especially since they are only mentioned as existent in the main text. If they don't provide additional relevant information, I'd rather remove them (e.g. time series for the three additional stations listed in the lower part of Table 1). If some of them show very specific features worth discussing, include them in the main manuscript.

Except for the first paragraph, the Conclusions are rather an Outlook with suggestions for future studies. It would be nice to have one or two more actual Conclusion points (which may require additional analysis).

General note on figures/figure captions: I could only print the manuscript in grey-scale. Many of the described features are barely or not at all visible when printed on grey-scale, i.e. not visible for color-blind people either. Please try some other color scales and also avoid references to color in the text/figure captions. In most cases, the color references can just be removed. Figures that caused me trouble in grey-scale are: Fig 1a,b (basically no spatial differences visible); Fig. 3 (colors of transects not distinguishable); Fig. 4 (colored regions not at all distinguishable from each other and from land; transect colors not distinguishable; remove color references in caption); Figs. 5 and 6 (just remove color references); Fig. 7a (gradient in darker colors not visible); Fig. 11 (gradients in darker colors barely visible: TN looks almost all the same throughout the year; locations of station hard to see)

**Specific comments**

Title: I think the speciality of the applied tracing approach is its quantitative nature. Therefore, I'd suggest a slight rephrasing of the title: "Quantifying the contribution of shipping NOx emissions to the marine nitrogen inventory – A case study for the western Baltic Sea".

Line 11: Please state why it is reduced during cyanobacteria blooms.

Lines 27/28: Is the difference in deposition velocity really the most important factor for land-sea differences? Isn't the spatial distribution of sources (over land >> over water) more important? You also mention it in the next sentence.

Lines 61-64: I would rephrase this part and get rid of the bullet point with the question, especially since it is only one question. E.g.: "Here, we combine such model with the nutrient tagging to quantify the contribution of shipping related nitrogen deposition to the total nitrogen (TN) and the different inorganic and organic nitrogen fractions."

Lines 69-72: I would move the part on Raudsepp et al (2013) to the discussion and discuss why one approach is better than the other depending on the research question. The tagging approach is preferable if the current state of a system is to be described as it doesn't change the balance

between sources ("non-disruptive approach"; see Menesguen and Lacroix (2018), doi: 10.1016/j.scitotenv.2018.04.183). However, if the effect of nutrient reductions should be determined, the actual removal of the considered source is required.

Lines 72-75: This should go right after the first mention of the tagging method, i.e. after the sentence ending on line 61.

Line 88: In the Discussion, you should include if 2012 is an average or exceptional year in terms of environmental conditions, nutrient inputs from different (types of) sources etc., and how this may affect your results (in case it is somehow exceptional).

Figs. 1 and 2: Change their order or adapt in-text cross-referencing. Currently, Fig. 2 is referred to first (line 104).

Lines 1007-111: Based on the introduction, do I understand correctly that shipping emissions only contribute to NOx? If so, maybe you could explicitly state that here. It could be worthwhile to provide a number or even a figure panel (in Fig. 1) for how much of the total NOx deposition is from ships.

Lines 121-124: Should this go into section 2.2 "Marine modelling"?

Lines 126-131: I suggest to move this to the discussion and add a statement on if and how this may affect the study results.

Lines 137/138: Is it really "ice cover thickness" and "extent"? Do you mean "ice cover" (as a fraction of the grid cell area) and "ice thickness"?

Lines 140-156: This needs a little bit of reordering. The sentence on the river loads should go to the end of the paragraph and the first sentence should be merged with the one on ERGOM's development at the IOW.

Lines 146/147: Has ERGOM atmospheric P deposition included? I am not sure after reading "phosphate" and "atmospheric deposition" in the same sentence. Please clarify.

Lines 160-162 (and thereafter throughout the manuscript): I suggest to simply distinguish between N (without "all") and ship-borne N ("N\_ship"). For ratios, I would then simply write "TN\_ship/TN" etc. It makes it more legible and text descriptions less cumbersome. Further suggest to rephrase this sentence to: "... another variable containing only shipping-related nitrogen (subscript "ship"). Process rates for the latter ones are equal to the process rates for the original state variables scaled by the relative contribution of shipping N to the educts."

Please also add on what frequency model output has been stored.

Lines 190-195: I suggest removing the part on oxygen. It is no relevant for the study, and it seems to me that persistent hypoxia/anoxia in the deep basins of the Baltic Sea is mixed up with seasonal hypoxia in parts of the coastal zone.

Fig. 4: Could be merged with Fig. 3 or at least put as Fig. 3b (ideally with the map inset shown in Fig. 3a/now Fig. 3). There is a horizontal line a few pixels above the horizontal border between

"Belt Sea" and "Arkona Basin", which has the same color as the "Oresund". Please double-check that there is no error with the region mask in your analyses.

Lines 208/209: Please state why you picked the three stations that are shown. Do they represent specific regimes? E.g. coastal vs. offshore? Re-evaluate whether you really need the three stations shown in the supplement. If you want to make the validation more sound, then you should point out somewhere in the validation section that model-data agreement is good also at the other stations presented in the supplement. Remove the sentence on the stations that were ignored.

Line 213: Could you specify what bathymetry criteria are used. Perhaps you can add relevant isobaths to Fig. 4?

Line 219 and Figs. 5 and 6: I would suggest switching the order of the T and S columns as you first talk about T and then S.

Lines 222/223: The vertical mixing might be a point for the discussion.

Lines 225-227: These two sentences contradict each other. First "all is good", then nitrate is not.

Lines 239-248: Should this go into the discussion?

Lines 244-246: I don't understand this part. Please clarify what is supposed to happen but does not in the model, and how this affects the deposition.

Fig. 8, caption: The last sentence explains how percentiles (10%, 50%, 90%) were calculated for the ratios. However, following this approach, all ratios get the same weight, which may change the results especially during periods of strong short-term changes (e.g. during the spring bloom). I would suggest calculating daily (assuming daily model output), spatial integrals of mass (i.e. concentration times volume) of TN and TN\_ship. You can then calculate TN\_ship/TN ratios and use the TN mass as weights to calculate weighted percentiles. This way you ensure that short-term changes in concentrations and volume (ERGOM uses a free surface, right?) are accounted for correctly. Analogously, you can calculate the weighted percentiles for TN by dividing the daily time series of regions' TN mass by the daily time series of the region' volumes, which gives the spatially weighted-average TN concentration, and using the daily time series of the regions' volumes as weights. Analogously for the other variables.

Fig. 10: What is the cause for the strong peaks in the 90% percentiles of PON (and TN) in fall and winter at DMU547?

Line 292: 6% is not "very high"

Line 297: An introductory sentence why you show the vertically resolved plots would be nice.

Lines 303-313: I would suggest rewriting this text part as a "normal" paragraph. The bullets make it appear like a list of very important information but they are mostly a detailed description of the dynamics of DIN, DON and PON. If you prefer keeping it as a bullets, please correct the numbering ("3" occurs twice).

Fig. 11: Would it make sense to add a row for PON since it's referred to a lot on lines 303-313? Perhaps you could label the two transects in Figs. 3/4 (e.g. T1/T2 or S1/S2) and use the label in the figure caption. I was a bit confused about the term "profiles": do you have profile data available? If so, why didn't you use them for validation?

Lines 329-332: Please explain the cause of the difference between Oresund and Arkona Basin.

Section 3.5: Include references to figures, it's hard to remember what result derives from which figure. Discussion should be in the same order as the results, i.e. annual cycle (lines 329-332) should be discussed before the vertical. And as mentioned in my general comments, please provide an actual discussion.

**Technical corrections**

Line 8: "the atmospheric input" Line 9: "in shallow coastal regions" Line 19: Include reference to MSFD (EU, 2008) for the GES Line 24: remove "approximately" Line 25: "atmospheric nitrogen deposition" Line 32: "region where high amounts" Line 43: move HELCOM reference to end of sentence Line 45: "That means" Line 46: "ships built after 2021" Lines 56/57: "in the system and their deposition sites" Line 60: "key variables in biogeochemical models" Lines 63/64: You also provide analyses for organic N, not just TN and DIN Lines 67/68: remove the sentence "We used an ..." Line 68: "Within ERGOM, the tagging" instead of "Previously, this tagging" Line 71: "and another one without" Line 79: "were not coupled on-line." Line 98: "deposited"; use full name of SMOKE Line 99: use full name of STEAM Line 100: use full term for IMO Caption of Fig. 1: remove long name of CMAQ and the note Line 109: Please add in parentheses what HONO is Line 138: remove "past" Line 146: "Inorganic" instead of "Basic"? Line 155/156: remove the name of the supplement; it can all be found in the readme. Line 157: "by the method" Line 158: "and used" Line 173: "in the eastern" Lines 178-180: change order of the two sentences Line 181: "emergence of stratification" Line 183: remove "read" Line 185: "N:P" => N and P have not been introduced Lines 189/190: "algal bloom period ends in autumn when stratification" Line 193/194: remove "- denoted as oxygen minimum zones -" Lines 205/206: "were aggregated into a monthly 'climatology'" ('climatology' with apostrophes as seven years do not really warrant a proper climatology)

Table 1, caption: remove "and model evaluation"; remove color references to figures

Table 1 should come before Fig. 4 as it is referred to first.

Fig. 5: remove "var [unit] (see header)" on the left; Why are some individual ticks on the y axes not labelled (e.g. 0 in bottom left panel or 15 in the one next to it)? Use "mmol N m-3" and "mmol P m-3" as units for NO3 and PO4, respectively; caption: remove color references; "Each column presents one state variable"; "Vertical lines and shaded areas show the monthly"; remove everything after the first "Supplement"

Line 212: "Basin definitions by Omstedt"

Line 215: "the model's open boundary"

Line 219: "Sea surface temperature [...] but sea surface salinity"

Fig. 6: remove "var [unit] (see header)" on the left; Why are some individual ticks on the y axes not labelled (e.g. 15 in bottom left panel)? Use "mmol N m-3" and "mmol P m-3" as units for NO3 and PO4, respectively; Why are the depth ranges at the first 2 stations not equivalent to 10m as stated on line 198? Caption: "Same as Fig. 5 but for the bottom 10 m."

Line 222: Here and throughout the manuscript, do you mean "seasonal" with "intra-annual"? Since you show monthly data, any shorter-term variability is averaged out.

Lines 227/228: replace "The modeled water column is stronger stratified than the real water column" with "Simulated salinity suggests that stratification is overestimated by the model"

Fig. 7: Should the unit be "mmol m-3", not "µmol m-3"? Remove the "all" subscript; stations/transects hard to see in grey-scale; caption: remove color references; "White areas are on land in the MOM-ERGOM domain."

Lines 239-241: "origin and caused by a lower horizontal grid resolution of the CMAQcompared to MOM-ERGOM and interpolation over the land-sea interface."; remove the last sentence Line 247: remove "to the ground/sea"

Line 249: here and later: "The contribution of shipping-related nitrogen to TN (TN\_ship/TN) ..."

Fig. 8: I would suggest removing the absolute shipping TN, DIN, etc. since it is pretty much zero compared to the overall quantities and hardly visible especially in grey-scale. The y axes labels could then be "overall" for the even rows and "shipping"; I suggest putting TN in the first column as you previously analyzed TN, so now you increase the level of detail from TN to its inorganic and organic compartments; add to the caption that it's 2012 only and remove "in 2012" from the x axes labels; caption: "thick" instead of "think"; remove "in the odd rows"; "For the ratios, …"; remove color references

Line 255: remove "with all nitrogen" and "all" subscript

Lines 259/260: "which are the sum of DIN, DON and PON"

Line 261: "In contrast, the DIN concentrations are elevated (~5 mmol N m-3) throughout the year in the Oresund ."

Lines 262/263: "by riverine nutrient loads"; remove the names of the rivers listed in parentheses

Lines 264/265: "The relative contributions of shipping N to DIN, DON and PON are very small."

Lines 267/268: "from 1.5-2% in January to about 1% in July"

Fig. 9: These are no daily values (see caption); put only one station/depth label per station (like in Fig. 8: change caption to: "Same as Fig. 8 but for specific stations (see Fig. 1)."

Line 274: "However" instead of "But"

Line 275: remove "as presented in Sect. 3.2"; "in the open ocean"

Line 278: These are no means and no daily data

Fig. 10: put only one station/depth label per station (like in Fig. 8); again why I sthe depth range not equal to 10 m for the first two stations? Change caption to: "Same as Fig. 9 but for the bottom 10 m."

Line 286: remove "data"

Line 291: "at the surface due to vertical stratification."

Line 297: State what quantities are shown.

Line 299: "causing the low values in the south."

Line 300: "later" instead of "delayed"

Line 303: remove "reaches ~12.5%"

Fig. 11: add "latitude (N)" to x axis on left and right; station lines are barely visible in grey scale, same for the temporal development; caption: remove "is plotted", remove last sentence

Line 315: Please rewrite the sentence such that models and data are only mentioned once

Line 317: "The concentration of shipping-related TN ..."

Line 320: "... the contribution of shipping-related N to DIN was highest ..."

Line 334: "to TN"

Line 335: "the NOx emissions"

Line 337: "to DIN"

Line 349: "." after ")"

Line 368: add first name of co-author (although I would suggest to only use initials in the whole author contribution section)

Line 383: "program whose intense"

---

## Short Comment (SC1) · 26 Aug 2019

Dear Fabian,

Thank you for the prompt and detailed review of the manuscript. Originally, I planned to reply to all reviewers and submit the revised manuscript at once. Since the second review (and finding a second reviewer at all) took/takes some time,

Unfortunately, we did not write out all processes. Therefore, we cannot do the full budget calculations and change "budget" to "inventory". We will include a comparison to other published studies in the discussion and try to add a clear "red line" to the

[Figure]

Results & Discussion section.

Thanks also for the remark to the choice of colors in the plots (with respect to printing black-only and color blind people). We used already the Viridis color scale in the spatial plots to account for this. However, we did not consider this aspect in the line plots. We will optimized them as well.

Daniel

---

## Referee Comment (RC2) · Anonymous Referee #2 · 3 Sep 2019

General comments:

The manuscript attempts to answer a relevant ocean research question, which is very clearly stated in the title.

The manuscript reads in a clear, concise, and well-structured way. The scientific approach is transparent and the methods and results are presented in an appropriate way.

However, the manuscript is lacking in the discussion and conclusions sections.

Specific comments:

[Figure]

The summarizing discussion section mainly consists of a summary of the results and very little actual discussion and the results are not set in context to relative literature.

The conclusions section partly consists of discussion and recommendations for further studies. It is not very clear what the conclusions are, except ". . . , the shipping sector might relevantly contribute to eutrophication at specific locations in the wester Baltic Sea in summer."

It seems too unsubstantial for the work that has been done and needs to be improved.

Technical corrections:

Page 2, Line 33: I think it should be 'where' instead of 'with'.

Page 3, Line 79: Remove 'But' at the start of the sentence.

Page 16, Line 88: Consider rephrasing ". . . stations distant to the coast . . .".

---

## Author Comment (AC2) · 20 Nov 2019

**Response to the comments of Reviewer #1**

We thank Fabian Große for the constructive comments on the manuscript. His comments are written in **bold font**. The authors' replies start with a ">" and are written in normal font.

**1 General comments**

The "Summarizing Discussion" is less of a discussion but more of a summary. Results should be discussed in the context of existing literature (e.g. comparison with the approach of Raudsepp et al. (2013) mentioned in the Introduction; is the tagging approach better/preferable? Why?). It needs to be very clear what the main insights and contributions of the present study are, and how they expand on previous knowledge. In addition, the limitations of the study need to be discussed. For instance, does the year 2012 represent an average year in terms of environmental conditions, inputs from different types of nutrient sources etc. or was it an exception with respect to some factors? If the latter is the case, what implications could that have on the generality of the results. Some limitations of the model are touched in the course of the Results section but those get a bit lost and they are not sufficient for an in-depth discussion. The authors further emphasize the role of the sediment in the Conclusions. So, what influence of the relatively simple sediment parameterization can be expected with respect to the presented results? (Note that this list of potential discussion points is not meant to be exhaustive.) The authors may want to consider separating Results from Discussion more clearly.

As mentioned above, an actual TN budget for overall and ship-borne N (e.g. for the basins defined in Fig. 4) could be a very nice addition to the manuscript. Or even a budget of the overall and shipborne N fluxes between the different model state variables (similar to a model schematic with numbers for the different fluxes for overall and ship-borne N). Since the manuscript is quite concise in its current form, this would not make it overly lengthy. If it's not possible to calculate such budget, i.e. the required fluxes were not stored, the authors should consider changing the title by using "inventory" instead of "budget".

The Results section is a bit of a laundry list without a clear transitions between
the individual results describing why the specific (upcoming) results are shown. Including such transitions from one subsection/result to the next would help the reader to get what the key focus of each figure is. Also, as a general comment on the results and the discussion: please regularly include crossreferences to figures and figure panels to allow for an easier link between text description and figures. You may want to add panel labels (a, b, c, ...) for easier in-text referencing.

I don't think all additional figures provided in the supplement are necessary, especially since they are only mentioned as existent in the main text. If they don't provide additional relevant information, I'd rather remove them (e.g. time series for the three additional stations listed in the lower part of Table 1). If some of them show very specific features worth discussing, include them in the main manuscript.

Except for the first paragraph, the Conclusions are rather an Outlook with suggestions for future studies. It would be nice to have one or two more actual Conclusion points (which may require additional analysis).

> We split the "*Results and Discussions*" section into two separate "*Results*" and "*Discussion*" sections to better differentiate between results and interpretation. Additional references to the figures were added. We updated the "*Conclusions*" section. Replies to further aspects mentioned in the previous five paragraphs of the reviewer's comment are part of replies to other reviewer's comments further below.

General note on figures/figure captions: I could only print the manuscript in grey-scale. Many of the described features are barely or not at all visible when printed on grey-scale, i.e. not visible for color-blind people either. Please try some other color scales and also avoid references to color in the text/figure captions. In most cases, the color references can just be removed. Figures that caused me trouble in grey-scale are: Fig 1a,b (basically no spatial differ-
ences visible); Fig. 3 (colors of transects not distinguishable); Fig. 4 (colored regions not at all distinguishable from each other and from land; transect colors not distinguishable; remove color references in caption); Figs. 5 and 6 (just remove color references); Fig. 7a (gradient in darker colors not visible); Figs. 8-10 (color references need to be removed, perhaps different line styles do the trick?); Fig. 11 (gradients in darker colors barely visible: TN looks almost all the same throughout the year; locations of station hard to see).

> Fig. 3: one transect is printed as dashed line, now; the colors are still mentioned in the legend but the descriptions are extended by further information for gray-scaled prints and color blind readers; basin names written in italics so that *land* and *basins* are distinguishable when printed in grayscales; isolines for bathymetry included in the plot;

> Fig. 4: changed colors of the masked regions; added think black contour to coastline to make land and masks distinguishable; one transect is printed as dashed line, now; the colors are still mentioned in the legend but the descriptions are extended by further information for gray-scaled prints and color blind readers; isolines for bathymetry included in the plot;

> Fig. 5/6: added a proper legend; removed reference to colors from the caption; observations are printer in dark red now (instead of *normal* red);

> Fig. 7a: unit corrected; we see no possibility to improve the color scale because it is already viridis; one cross section is dashed, now;

> Fig. 8-10: kept references to colors but added information on whether the color is lighter/darker than the other one: "green, lighter color in grayscales" and "blue, darker color in grayscales"

OSD
> Fig. 11: refined the color scale; moved symbol indicating the station's location further upwards; further modifications based on further reviewer comments further below

**2 Specific comments**

Title: I think the speciality of the applied tracing approach is its quantitative nature. Therefore, I'd suggest a slight rephrasing of the title: "Quantifying the contribution of shipping NOx emissions to the marine nitrogen inventory – A case study for the western Baltic Sea".

> modified title as suggested; thanks

**Line 11: Please state why it is reduced during cyanobacteria blooms.**

> We appended to the sentence "... because the cyanobacteria fix molecular nitrogen.".

Lines 27/28: Is the difference in deposition velocity really the most important factor for land-sea differences? Isn't the spatial distribution of sources (over land » over water) more important? You also mention it in the next sentence.

> Both aspects are important. There is a clear gradient from the coastline towards the open sea caused by the spatial distribution of emissions sources (higher emissions on land). However, the difference in the deposition velocity has the highest impact (e.g. Fig. 4a in Karl et al., 2019, doi: https://doi.org/10.5194/acp-19-1721-2019).
Lines 61-64: I would rephrase this part and get rid of the bullet point with the question, especially since it is only one question. E.g.: "Here, we combine such model with the nutrient tagging to quantify the contribution of shipping related nitrogen deposition to the total nitrogen (TN) and the different inorganic and organic nitrogen fractions."

> removed itemizations; slightly modified the question; kept the question as a question because some readers like to have an explicit research question in the introduction;

Lines 69-72: I would move the part on Raudsepp et al (2013) to the discussion and discuss why one approach is better than the other depending on the research question. The tagging approach is preferable if the current state of a system is to be described as it doesn't change the balance between sources ("*non-disruptive approach*"; see Menesguen and Lacroix (2018), doi: 10.1016/j.scitotenv.2018.04.183). However, if the effect of nutrient reductions should be determined, the actual removal of the considered source is required.

> We did not move the reference to Raudsepp et al. (2013) but extended the discussion on the mentioned topic (*"Why the choosen approach and not simulation without shipping-related nitrogen"*.

Lines 72-75: This should go right after the first mention of the tagging method, i.e. after the sentence ending on line 61.

> We prefer the order: problem description, research question, and brief presentation of the methods. Therefore, we removed the sentence "*Using a nutrient source tagging approach (e.g., Ménesguen et al., 2006)*..." (lines 59-61), in which the tagging method was mentioned frist, and kept the lines 72-75 where they were.
Line 88: In the Discussion, you should include if 2012 is an average or exceptional year in terms of environmental conditions, nutrient inputs from different (types of) sources etc., and how this may affect your results (in case it is somehow exceptional).

> a brief summary of the requested information:

- There were no exceptionally strong Baltic Sea inflows from the North Sea, which might have affected salinity, temperatures and other parameters (Mohrholz, 2018a).
- The EMEP Status Report 2014 compares the atmospheric conditions of the year 2012 with the conditions of previous twelve years (EMEP, 2014). Although neither the EMEP meteorological forcing (ECMWF IFS) nor the same EMEP emission data were used in this study, the descriptions in EMEP (2014) are valid for most aspects of the atmospheric forcing of this study for two reasons: (a) The meteorological forcing of this study (coastDat2 and coastDat3) is a reanalysis for which observational data were assimilated. Therefore, we can expect that the general meteorological features in the coastDat2/3 datasets are similar to those in the ECMWF IFS dataset. (b) The SMOKE for Europe emissions used in this study are largely based on the EMEP emissions. The spatio-temporal patterns of the emissions of several air pollutants differ but the annual sums are equal. One exception are the emissions of ammonia, which are calculated bottom-up in SMOKE for Europe in some European countries.
- The precipitation amount in Northern Europe in 2012 was above the long term average (EMEP, 2014; p.23).
- The nitrogen wet deposition in Northern Europe in 2012 was above the average of the previous ten years due to increased precipition (EMEP, 2014; p.49).
- The nitrogen dry deposition in Northern Europe in 2012 was lower than in the previous ten years (high wet and lower was deposition) but the total nitrogen deposion (dry + wet) was still higher (EMEP, 2014; p.49).
- The NOX emissions in 2012 were lower than in the previos ten year average leading to reduced nitrogen deposition on European average (EMEP, 2014; p.49).
- The increase in nitrogen deposition in Northern Europe in 2012 (due to strong wet deposition) was weakened by the lower NOX emissions (EMEP, 2014; p.49).
- The ammonia emissions are treated differently in SMOKE for Europe than in the EMEP emission model. Therefore, the information on reduced nitrogen deposition in EMEP (2014) is not applicable here. Unfortunately, the Emissions by SMOKE for Europe were specifically created for the year 2012 and are not fully comparable to previously by SMOKE for Europe create emissions of other years.

> Added this information as subsection to Sect. 2 (Materials and Methods)

Figs. 1 and 2: Change their order or adapt in-text cross-referencing. Currently, Fig. 2 is referred to first (line 104).

> changed order

Lines 107-111: Based on the introduction, do I understand correctly that shipping emissions only contribute to NOx? If so, maybe you could explicitly state that here. It could be worthwhile to provide a number or even a figure panel (in Fig. 1) for how much of the total NOx deposition is from ships.

> yes, corret, only NOx; included in the text; good idea; added another row of plots to the figure

OSD
**Lines 121-124: Should this go into section 2.2 "Marine modelling"?**

> Should be in the beginning of 2.1. Moved it there and modified text.

**Lines 126-131: I suggest to move this to the discussion and add a statement on if and how this may affect the study results.**

> We kept the sentence there but mention und discuss it in the discussion.

**Lines 137/138: Is it really "*ice cover thickness*" and "*extent*"? Do you mean "*ice cover*" (as a fraction of the grid cell area) and "*ice thickness*"?**

> forgot a comma; added information in brackets; now: "... simulates ice cover (fraction of grid cell area), thickness and extent"

Lines 140-156: This needs a little bit of reordering. The sentence on the river loads should go to the end of the paragraph and the first sentence should be merged with the one on ERGOM's development at the IOW.

> reordered and merged as suggested

Lines 146/147: Has ERGOM atmospheric P deposition included? I am not sure after reading "*phosphate*" and "*atmospheric deposition*" in the same sentence. Please clarify.

> yes, P deposition is included (as phosphate); Replaced "*or*" by "*and*" at two locations and added "*ammonium*" to reduce ambiguity; now: "*Inorganic nutrients – i.e. nitrate*
 $(NO_3^-, ammonium (NH_4^+), and phosphate (PO_4^{3-}) - enter the system via river input, atmospheric deposition, and remineralization of organic matter."$

Lines 160-162 (and thereafter throughout the manuscript): I suggest to simply distinguish between N (without "*all*") and ship-borne N ("Nship"). For ratios, I would then simply write "TNship/*TN*" etc. It makes it more legible and text descriptions less cumbersome. Further suggest to rephrase this sentence to: "...another variable containing only shipping-related nitrogen (subscript "ship"). Process rates for the latter ones are equal to the process rates for the original state variables scaled by the relative contribution of shipping N to the educts." Please also add on what frequency model output has been stored.

> We removed *all* at some locations. However, in our opinion the usage of *all* prevents ambiguity in some situations for some readers. Therefore, we prefer to keep it at most occasions.

> State variables in full spatial resolution were written out as monthly means. We had a daily output interval only at the locations of measurement stations. We added this information as an extra paragraph to the Materials and Methods Section on the marine model (currently Sect. 2.2).

Lines 190-195: I suggest removing the part on oxygen. It is no relevant for the study, and it seems to me that persistent hypoxia/anoxia in the deep basins of the Baltic Sea is mixed up with seasonal hypoxia in parts of the coastal zone.

> removed

Fig. 4: Could be merged with Fig. 3 or at least put as Fig. 3b (ideally with the map inset shown in Fig. 3a/now Fig. 3). There is a horizontal line a few pixels
above the horizontal border between "*Belt Sea*" and "*Arkona Basin*", which has the same color as the "*Oresund*". Please double-check that there is no error with the region mask in your analyses.

> We would prefer to keep the figures separated so that they are closer to the relevant text passages. The figures are planned as one-column figures in the final document. Hence, they will consume less space in the final document compared to the discussion version. Thanks for looking in details into the plot and identifying the horizontal line. The plotting programm bilinearly interpolated between the masks. We manually draw colored boxes atop of the interpolated regions, which we did not do properly. The issue is fixed now. It only affected this figure and no data.

Lines 208/209: Please state why you picked the three stations that are shown. Do they represent specific regimes? E.g. coastal vs. offshore? Re-evaluate whether you really need the three stations shown in the supplement. If you want to make the validation more sound, then you should point out somewhere in the validation section that model-data agreement is good also at the other stations presented in the supplement. Remove the sentence on the stations that were ignored.

> add text after first sentence: "They represent different regimes in the considered region: two offshore stations in different basins (OMBMPM2 and BY2) and one station close to the shore (DMU547). Additionally, sufficient measurement data were available at these locations.". Replaced the second sentence by "Validation plots at three additional stations are presented in the supplement and show a similar outcome. They are included to indicate that not just the three best stations are shown here.".

Line 213: Could you specify what bathymetry criteria are used. Perhaps you can add relevant isobaths to Fig. 4?
> added isolines for bathymetry to Fig. 4; the rough spatial extend of the domains/masks was taken from the referenced publication and then fine-tuned based on the used bathymetry

Line 219 and Figs. 5 and 6: I would suggest switching the order of the T and S columns as you first talk about T and then S.

> order of columns switched as suggested

Lines 222/223: The vertical mixing might be a point for the discussion.

> Included in paragraph *Discussion* (first paragraph).

Lines 225-227: These two sentences contradict each other. First "*all is good*", then nitrate is not.

> The one sentence ("*all is good*") means the sea surface and the other sentences means the sea floor. Add "... *at the sea floor*" to the first sentence to clarify this.

Lines 239-248: Should this go into the discussion?

> moved to the discussion

Lines 244-246: I don't understand this part. Please clarify what is supposed to happen but does not in the model, and how this affects the deposition.

> extended this passage and added reaction equations (now in the Sect. *Summarizing discussion*)
Fig. 8, caption: The last sentence explains how percentiles (10%, 50%, 90%) were calculated for the ratios. However, following this approach, all ratios get the same weight, which may change the results especially during periods of strong short-term changes (e.g. during the spring bloom). I would suggest calculating daily (assuming daily model output), spatial integrals of mass (i.e. concentration times volume) of TN and  $TN_{ship}$ . You can then calculate  $TN_{ship}/TN$  ratios and use the TN mass as weights to calculate weighted percentiles. This way you ensure that short-term changes in concentrations and volume (ERGOM uses a free surface, right?) are accounted for correctly. Analogously, you can calculate the weighted percentiles for TN by dividing the daily time series of regions' TN mass by the daily time series of the region' volumes, which gives the spatially weighted-average TN concentration, and using the daily time series of the regions' volumes as weights. Analogously for the other variables.

> For the basin means, we had only monthly mean values available as basis for the calculations. Temporally higher resolved data were only written out at the locations of measurement stations. This was done to save disc space. The plotted variability rather is a spatial variability than as temporal one. We added this information to the figure's caption.

**Fig. 10: What is the cause for the strong peaks in the 90% percentiles of PON (and TN) in fall and winter at DMU547?**

> The detritus concentrations are very high at three days in September and December. The high concentrations correlate with peaks in the U and V current velocities at the sea floor. Probably, detritus was resuspended from the sea floor. Values of relevant state variables at different depths at this station are, now, provided in the supplement (dmu547\_peak\_autumn.pdf);
Line 292: 6% is not "very high"

> replaced by "... ratio peaks with  $\approx 6\%$  ... "

Line 297: An introductory sentence why you show the vertically resolved plots would be nice.

> added two introductory sentences

Lines 303-313: I would suggest rewriting this text part as a "*normal*" paragraph. The bullets make it appear like a list of very important information but they are mostly a detailed description of the dynamics of DIN, DON and PON. If you prefer keeping it as a bullets, please correct the numbering ("*3*" occurs twice).

> itemization converted into one paragraph

Fig. 11: Would it make sense to add a row for PON since it's referred to a lot on lines 303-313? Perhaps you could label the two transects in Figs. 3/4 (e.g. T1/T2 or S1/S2) and use the label in the figure caption. I was a bit confused about the term "*profiles*": do you have profile data available? If so, why didn't you use them for validation?

> added T1/T2 to Figs. 3 and 4; added PON to Fig. 11; We have profile data available and we used them to calculate the 10 m-averages in the validation section. However, it was not mentioned explicitly in the *Materials & Methods* section. We added a sentence "*Vertical profiles were measured at most stations.*" to the MM Section.
Lines 329-332: Please explain the cause of the difference between Oresund and Arkona Basin. Section 3.5: Include references to figures, it's hard to remember what result derives from which figure. Discussion should be in the same order as the results, i.e. annual cycle (lines 329-332) should be discussed before the vertical. And as mentioned in my general comments, please provide an actual discussion.

> added explanation (in the Discussion)

- > references to the figures added
- > moved paragraph on vertical distribution further downward
- 3 Technical corrections
- Line 8: "the atmospheric input"
- > included
- Line 9: "in shallow coastal regions"

> replaced

Line 19: Include reference to MSFD (EU, 2008) for the GES

> included
**Line 24: remove "approximately"**

> removed

Line 25: "atmospheric nitrogen deposition"

> added

Line 32: "region where high amounts"

> modified

Line 43: move HELCOM reference to end of sentence

> moved

Line 45: "That means"

> modified

Line 46: "ships built after 2021"

> modified

Lines 56/57: "in the system and their deposition sites"
> modified; but used "the location of their deposition sites" instead of "their deposition sites"

Line 60: "key variables in biogeochemical models"

> modified

Lines 63/64: You also provide analyses for organic N, not just TN and DIN

> Thanks. We will include it.

Lines 67/68: remove the sentence "We used an ... "

> removed

Line 68: "Within ERGOM, the tagging" instead of "Previously, this tagging"

> modified

Line 71: "and another one without"

> modified

Line 79: "were not coupled on-line."

> modified
> modified; included

**Line 99: use full name of STEAM**

> included

**Line 100: use full term for IMO**

> included

**Caption of Fig. 1: remove long name of CMAQ and the note**

> removed

**Line 109: Please add in parentheses what HONO is**

> HONO is no abbreviation of a substance name like PAN and PNA but the chemical formula. We reordered the list to have the two abbreviations in the end.

Line 138: remove "past"

> removed

```
Line 146: "Inorganic" instead of "Basic"?
```
> replaced

Line 155/156: remove the name of the supplement; it can all be found in the readme.

> removed

Line 157: "by the method"

> modified

Line 158: "and used"

> modified

Line 173: "in the eastern"

> modified

Lines 178-180: change order of the two sentences

> changed order

Line 181: "emergence of stratification"

> modified
**Line 183: remove "read"**

> removed

Line 185: "N:P" => N and P have not been introduced

> added "nitrogen-to-phosphorus"

Lines 189/190: "algal bloom period ends in autumn when stratification"

> modified

Line 193/194: remove "- denoted as oxygen minimum zones -"

> removed

Lines 205/206: *"were aggregated into a monthly 'climatology"* ('climatology' with apostrophes as seven years do not really warrant a proper climatology)

> added

Table 1, caption: remove "and model evaluation"; remove color references to figures

> removed; reformulated caption slightly

Table 1 should come before Fig. 4 as it is referred to first.
> Table 1 is before Fig. 4 in the LaTeX document. We will take care that they will be properly ordered in the final version.

Fig. 5: remove "*var [unit] (see header)*" on the left; Why are some individual ticks on the y axes not labelled (e.g. 0 in bottom left panel or 15 in the one next to it)? Use "*mmol N m-3*" and "*mmol P m-3*" as units for NO3 and PO4, respectively; caption: remove color references; "*Each column presents one state variable*"; "*Vertical lines and shaded areas show the monthly*"; remove everything after the first "*Supplement*"

> removed "var [unit] (see header)"; Default plotting in R – better this way (adjusted minimum of y-axis; decreased font size of two plots; manually added some values along the y-axis)? Otherwise the labels would have a too small font size; We would like to keep the units SI conformal but 'N' and 'P' are not defined by SI. As long was we have amount ('moles') and not mass ('g'), *x* mol of N are equal to *x* mol of nitrate; colored references removed; column/row corrected; "*the*" in the sentence on "*vertical lines and shaded areas*" removed; removed text after "*Supplement*";

Line 212: "Basin definitions by Omstedt"

> modified

Line 215: "the model's open boundary"

> modified

Line 219: "Sea surface temperature [...] but sea surface salinity"
Fig. 6: remove "var [unit] (see header)" on the left; Why are some individual ticks on the y axes not labelled (e.g. 15 in bottom left panel)? Use "mmol N m-3" and "mmol P m-3" as units for NO3 and PO4, respectively; Why are the depth ranges at the first 2 stations not equivalent to 10m as stated on line 198? Caption: "Same as Fig. 5 but for the bottom 10 m."

> adapted like in Fig.5; In shallow regions below  $\approx 30$  m less than 10 m (above the bottom) were considered. Statement in I. 198 was correct; first sentence simplified similar as suggested

Line 222: Here and throughout the manuscript, do you mean "*seasonal*" with "*intra-annual*"? Since you show monthly data, any shorter-term variability is averaged out.

> Yes, we meant "seasonal". We replaced "intra-annual" by "seasonal".

Lines 227/228: replace "The modeled water column is stronger stratified than the real water column" with "Simulated salinity suggests that stratification is overestimated by the model"

> replaced

Fig. 7: Should the unit be "*mmol m-3*", not " $\mu$ *mol m-3*"? Remove the "all" subscript; stations/transects hard to see in grey-scale; caption: remove color references; "White areas are on land in the MOM-ERGOM domain."
> Yes, should be "*mmol m-3*"; We think that it is a good idea to add a subscript to all TN to be unambiguous; removed color references;

Lines 239-241: "origin and caused by a lower horizontal grid resolution of the CMAQcompared to MOM-ERGOM and interpolation over the land-sea interface."; remove the last sentence

> modified

Line 247: remove "to the ground/sea"

> removed

Line 249: here and later: "The contribution of shipping-related nitrogen to TN  $(TN_{ship}/TN)$ ..."

> resolved here but not at later locations

Fig. 8: I would suggest removing the absolute shipping TN, DIN, etc. since it is pretty much zero compared to the overall quantities and hardly visible especially in grey-scale. The y axes labels could then be "*overall*" for the even rows and "*shipping*"; I suggest putting TN in the first column as you previously analyzed TN, so now you increase the level of detail from TN to its inorganic and organic compartments; add to the caption that it's 2012 only and remove "*in 2012*" from the x axes labels; caption: "*thick*" instead of "*think*"; remove "*in the odd rows*"; "*For the ratios, …*"; remove color references

OSD
> We would like to keep the green lines in odd rows because they make hugh difference between untagged nitrogen and shipping-related nitrogen clear and do not disturb the understanding of the plots.

> order of DIN, DON, PON, and TN: DIN is described first and TN last in Sect. 3.3. Hence, it is reasonable to keep the order as it is. For the same reason, the *temperature* and *salinity* were switched in Figs. 5 and 6.

> replaced "think" by "thick"

> kept color reference but added description in terms of darker/lighter color

Line 255: remove "with all nitrogen" and "all" subscript

> removed "with all nitrogen" but kept the subscript to prevent ambiguity

Lines 259/260: "which are the sum of DIN, DON and PON"

> modified

Line 261: "In contrast, the DIN concentrations are elevated ( $\approx 5 \text{ mmol N m}^{-3}$ ) throughout the year in the Oresund ."

> modified

Lines 262/263: "*by riverine nutrient loads*"; remove the names of the rivers listed in parentheses

OSD
> removed

Lines 264/265: "The relative contributions of shipping N to DIN, DON and PON are very small."

> modified

Lines 267/268: "from 1.5-2% in January to about 1% in July"

> modified

Fig. 9: These are no daily values (see caption); put only one station/depth label per station (like in Fig. 8: change caption to: "Same as Fig. 8 but for specific stations (see Fig. 1)."

> The sentence was formulated ambiguously. We have daily mean values and calculate monthly percentiles from them. In Fig. 8, we had only monthly mean values. Background: We saved model output at specific locations (measurement stations) in daily resolution. But, the full spatial model output was only stored in monthly resolution to space disc space. Fig. 9 shows the variability in time, whereas Fig.8 shows variability in space.

Line 274: "However" instead of "But"

> modified

Line 275: remove "as presented in Sect. 3.2"; "in the open ocean" C25
> removed and modified

Line 278: These are no means and no daily data

> adapted to be less ambiguous; see reply to comment on Fig. 9;

Fig. 10: put only one station/depth label per station (like in Fig. 8); again why I sthe depth range not equal to 10 m for the first two stations? Change caption to: *"Same as Fig. 9 but for the bottom 10 m."*

> station label: updated; depth range: see reply to comment on Fig. 6; modified caption similar as suggested;

Line 286: remove "data"

> removed

Line 291: "at the surface due to vertical stratification."

> modified

Line 297: State what quantities are shown.

> added; also included information on temporal resolution

Line 299: "causing the low values in the south."
> modified

Line 300: "later" instead of "delayed"

> modified

Line 303: remove "reaches  $\approx 12.5$  %"

> removed

Fig. 11: add "*latitude (N)*" to x axis on left and right; station lines are barely visible in grey scale, same for the temporal development; caption: remove "*is plotted*", remove last sentence

> added 'latitude (N)'; changed color of station-symbol and -line; improvement of colorscale does not seem to be possible when it should remain equal for all 24 plots; removed both;

Line 315: Please rewrite the sentence such that models and data are only mentioned once

> rewritten

Line 317: "The concentration of shipping-related TN ...."

> modified
**Line 320: "... the contribution of shipping-related N to DIN was highest ... "**

> modified

Line 334: "to TN"

> modified

Line 335: "the NOx emissions"

> modified

Line 337: "to DIN"

> modified

Line 349: "." after ")"

> added

Line 368: add first name of co-author (although I would suggest to only use initials in the whole author contribution section)

> changed to: only initials

Line 383: "program whose intense"

OSD
> modified

---

## Author Response (AR1)

Daniel Neumann1, Matthias Karl2, Hagen Radtke1, Volker Matthias2, René Friedland3, and Thomas Neumann1

1Leibniz-Institute for Baltic Sea Research Warnemünde, Seestr. 15, 18119 Rostock, Germany
 2Institute of Coastal Research, Helmholtz-Zentrum Geesthacht, Max-Planck-Str. 1, 21502 Geesthacht, Germany
 3European Commission DG Joint Research Centre, Directorate D – Sustainable Resources, Via Fermi, 2749 – TP 270, I-21027 Ispra (VA), Italy

Correspondence: D. Neumann (daniel.neumann@io-warnemuende.de.de)

**Response to the comments of Reviewer #1**

We thank Fabian Große for the constructive comments on the manuscript. His comments are written in **bold font**. The authors' replies start with a ">" and are written in normal 5 font.

**1** General comments**

The "Summarizing Discussion" is less of a discussion but more of a summary. Results should be discussed in the context of existing literature (e.g. comparison with the 10 approach of Raudsepp et al. (2013) mentioned in the Introduction; is the tagging approach better/preferable? Why?). It needs to be very clear what the main insights and contributions of the present study are, and how they expand on previous knowledge. In addition, the limita-

- 15 tions of the study need to be discussed. For instance, does the year 2012 represent an average year in terms of environmental conditions, inputs from different types of nutrient sources etc. or was it an exception with respect to some factors? If the latter is the case, what implications
  20 could that have on the generality of the results. Some lim-
- itations of the model are touched in the course of the Results section but those get a bit lost and they are not sufficient for an in-depth discussion. The authors further emphasize the role of the sediment in the Conclusions.

25 So, what influence of the relatively simple sediment parameterization can be expected with respect to the presented results? (Note that this list of potential discussion points is not meant to be exhaustive.) The authors may want to consider separating Results from Discussion more clearly.

As mentioned above, an actual TN budget for overall and ship-borne N (e.g. for the basins defined in Fig. 4) could be a very nice addition to the manuscript. Or even a budget of the overall and shipborne N fluxes between the different model state variables (similar to a model schematic 35 with numbers for the different fluxes for overall and shipborne N). Since the manuscript is quite concise in its current form, this would not make it overly lengthy. If it's not possible to calculate such budget, i.e. the required fluxes were not stored, the authors should consider changing the 40 title by using "inventory" instead of "budget".

The Results section is a bit of a laundry list without a clear transitions between the individual results describing why the specific (upcoming) results are shown. Including such transitions from one subsection/result to the 45 next would help the reader to get what the key focus of each figure is. Also, as a general comment on the results and the discussion: please regularly include crossreferences to figures and figure panels to allow for an easier link between text description and figures. You may want 50 to add panel labels (a, b, c, ...) for easier in-text referencing.

I don't think all additional figures provided in the supplement are necessary, especially since they are only mentioned as existent in the main text. If they don't provide 55 additional relevant information, I'd rather remove them (e.g. time series for the three additional stations listed in the lower part of Table 1). If some of them show very specific features worth discussing, include them in the main manuscript.

5 Except for the first paragraph, the Conclusions are rather an Outlook with suggestions for future studies. It would be nice to have one or two more actual Conclusion points (which may require additional analysis).

> We split the "*Results and Discussions*" section into two 10 separate "*Results*" and "*Discussion*" sections to better differentiate between results and interpretation. Additional references to the figures were added. We updated the "*Conclusions*" section. Replies to further aspects mentioned in the previous five paragraphs of the reviewer's comment are part 15 of replies to other reviewer's comments further below.

General note on figures/figure captions: I could only print the manuscript in grey-scale. Many of the described features are barely or not at all visible when printed on grey-scale, i.e. not visible for color-blind people either.

- 20 Please try some other color scales and also avoid references to color in the text/figure captions. In most cases, the color references can just be removed. Figures that caused me trouble in grey-scale are: Fig 1a,b (basically no spatial differences visible); Fig. 3 (colors of transects not
- 25 distinguishable); Fig. 4 (colored regions not at all distinguishable from each other and from land; transect colors not distinguishable; remove color references in caption); Figs. 5 and 6 (just remove color references); Fig. 7a (gradient in darker colors not visible); Figs. 8-10 (color references)
- 30 ences need to be removed, perhaps different line styles do the trick?); Fig. 11 (gradients in darker colors barely visible: TN looks almost all the same throughout the year; locations of station hard to see).

> Fig. 3: one transect is printed as dashed line, now; the 35 colors are still mentioned in the legend but the descriptions are extended by further information for gray-scaled prints and color blind readers; basin names written in italics so that *land* and *basins* are distinguishable when printed in grayscales; isolines for bathymetry included in the plot;

- Fig. 4: changed colors of the masked regions; added think black contour to coastline to make land and masks distinguishable; one transect is printed as dashed line, now; the colors are still mentioned in the legend but the descriptions are extended by further information for gray-scaled prints 45 and color blind readers; isolines for bathymetry included in
- the plot;

> Fig. 5/6: added a proper legend; removed reference to colors from the caption; observations are printer in dark red now (instead of *normal* red);

50 > Fig. 7a: unit corrected; we see no possibility to improve the color scale because it is already viridis; one cross section is dashed, now; > Fig. 8-10: kept references to colors but added information on whether the color is lighter/darker than the other one: *"green, lighter color in grayscales"* and *"blue, darker color* 55 *in grayscales"*

> Fig. 11: refined the color scale; moved symbol indicating the station's location further upwards; further modifications based on further reviewer comments further below

**2 Specific comments**

Title: I think the speciality of the applied tracing approach is its quantitative nature. Therefore, I'd suggest a slight rephrasing of the title: "Quantifying the contribution of shipping NOx emissions to the marine nitrogen inventory – A case study for the western Baltic Sea".

> modified title as suggested; thanks

Line 11: Please state why it is reduced during cyanobacteria blooms.

> We appended to the sentence "... because the cyanobacteria fix molecular nitrogen.".

Lines 27/28: Is the difference in deposition velocity really the most important factor for land-sea differences? Isn't the spatial distribution of sources (over land » over water) more important? You also mention it in the next sentence.

> Both aspects are important. There is a clear gradient from the coastline towards the open sea caused by the spatial distribution of emissions sources (higher emissions on land). However, the difference in the deposition velocity has the highest impact (e.g. Fig. 4a in Karl et al., 2019, doi: 80 https://doi.org/10.5194/acp-19-1721-2019).

Lines 61-64: I would rephrase this part and get rid of the bullet point with the question, especially since it is only one question. E.g.: "Here, we combine such model with the nutrient tagging to quantify the contribution of shipping related nitrogen deposition to the total nitrogen (TN) and the different inorganic and organic nitrogen fractions."

> removed itemizations; slightly modified the question; kept the question as a question because some readers like to have an explicit research question in the introduction;

Lines 69-72: I would move the part on Raudsepp et al (2013) to the discussion and discuss why one approach is better than the other depending on the research question. The tagging approach is preferable if

60

65

70

75

90

the current state of a system is to be described as it doesn't change the balance between sources ("*nondisruptive approach*"; see Menesguen and Lacroix (2018), doi: 10.1016/j.scitotenv.2018.04.183). However, if the ef-5 fect of nutrient reductions should be determined, the actual removal of the considered source is required.

> We did not move the reference to Raudsepp et al. (2013) but extended the discussion on the mentioned topic ("Why the choosen approach and not simulation without shipping-10 related nitrogen".

**Lines 72-75: This should go right after the first mention of the tagging method, i.e. after the sentence ending on line 61.**

> We prefer the order: problem description, research ques-15 tion, and brief presentation of the methods. Therefore, we removed the sentence "*Using a nutrient source tagging approach (e.g., Ménesguen et al., 2006)* ..." (lines 59-61), in which the tagging method was mentioned frist, and kept the lines 72-75 where they were.

- 20 Line 88: In the Discussion, you should include if 2012 is an average or exceptional year in terms of environmental conditions, nutrient inputs from different (types of) sources etc., and how this may affect your results (in case it is somehow exceptional).
- 25 > a brief summary of the requested information:
  - There were no exceptionally strong Baltic Sea inflows from the North Sea, which might have affected salinity, temperatures and other parameters (Mohrholz, 2018a).
- The EMEP Status Report 2014 compares the atmospheric conditions of the year 2012 with the conditions of previous twelve years (EMEP, 2014). Although neither the EMEP meteorological forcing (ECMWF IFS) nor the same EMEP emission data were used in this study, the descriptions in EMEP (2014) are valid for most aspects of the atmospheric forcing of this study for two reasons: (a) The meteorological forcing of this study (coastDat2 and coastDat3) is a reanalysis for which observational data were assimilated. Therefore, we can expect that the general meteorological features
- in the *coastDat2/3* datasets are similar to those in the *ECMWF IFS* dataset. (b) The *SMOKE for Europe* emissions used in this study are largely based on the EMEP emissions. The spatio-temporal patterns of the emissions of several air pollutants differ but the annual sums are equal. One exception are the emissions of ammonia,
- which are calculated bottom-up in *SMOKE for Europe* in some European countries.
  - The precipitation amount in Northern Europe in 2012 was above the long term average (EMEP, 2014; p.23).

- The nitrogen wet deposition in Northern Europe in 2012 50 was above the average of the previous ten years due to increased precipition (EMEP, 2014; p.49).
- The nitrogen dry deposition in Northern Europe in 2012 was lower than in the previous ten years (high wet and lower was deposition) but the total nitrogen deposion 55 (dry + wet) was still higher (EMEP, 2014; p.49).
- The NOX emissions in 2012 were lower than in the previos ten year average leading to reduced nitrogen deposition on European average (EMEP, 2014; p.49).
- The increase in nitrogen deposition in Northern Europe 60 in 2012 (due to strong wet deposition) was weakened by the lower NOX emissions (EMEP, 2014; p.49).
- The ammonia emissions are treated differently in *SMOKE for Europe* than in the EMEP emission model. Therefore, the information on reduced nitrogen deposition in EMEP (2014) is not applicable here. Unfortunately, the Emissions by *SMOKE for Europe* were specifically created for the year 2012 and are not fully comparable to previously by *SMOKE for Europe* create emissions of other years.

> Added this information as subsection to Sect. 2 (*Materials and Methods*)

Figs. 1 and 2: Change their order or adapt in-text crossreferencing. Currently, Fig. 2 is referred to first (line 104).

75

> changed order

Lines 107-111: Based on the introduction, do I understand correctly that shipping emissions only contribute to NOx? If so, maybe you could explicitly state that here. It could be worthwhile to provide a number or even a figure panel (in Fig. 1) for how much of the total NOx deposition is from ships.

> yes, corret, only NOx; included in the text; good idea; added another row of plots to the figure

**Lines 121-124: Should this go into section 2.2 "Marine 85 modelling"?**

> Should be in the beginning of 2.1. Moved it there and modified text.

**Lines 126-131: I suggest to move this to the discussion and add a statement on if and how this may affect the study 90 results.**

> We kept the sentence there but mention und discuss it in the discussion.

Lines 137/138: Is it really *"ice cover thickness"* and *"extent"*? Do you mean *"ice cover"* (as a fraction of the grid cell area) and *"ice thickness"*?

> forgot a comma; added information in brackets; now: 5 "... simulates ice cover (fraction of grid cell area), thickness and extent"

Lines 140-156: This needs a little bit of reordering. The sentence on the river loads should go to the end of the paragraph and the first sentence should be merged with 10 the one on ERGOM's development at the IOW.

> reordered and merged as suggested

Lines 146/147: Has ERGOM atmospheric P deposition included? I am not sure after reading "*phosphate*" and "*atmospheric deposition*" in the same sentence. Please 15 clarify.

> yes, P deposition is included (as phosphate); Replaced "or" by "and" at two locations and added "ammonium" to reduce ambiguity; now: "Inorganic nutrients – i.e. nitrate (NO3-, ammonium (NH4+), and phosphate (PO43-) – enter
20 the system via river input, atmospheric deposition, and remineralization of organic matter."

Lines 160-162 (and thereafter throughout the manuscript): I suggest to simply distinguish between N (without "all") and ship-borne N ("Nship"). For ratios, 25 I would then simply write "TNship/TN" etc. It makes it more legible and text descriptions less cumbersome. Further suggest to rephrase this sentence to: "... another variable containing only shipping-related nitrogen (subscript "ship"). Process rates for the latter ones are equal 30 to the process rates for the original state variables scaled by the relative contribution of shipping N to the educts." Please also add on what frequency model output has been stored.

> We removed *all* at some locations. However, in our opin-35 ion the usage of *all* prevents ambiguity in some situations for some readers. Therefore, we prefer to keep it at most occasions.

> State variables in full spatial resolution were written out as monthly means. We had a daily output interval only at 40 the locations of measurement stations. We added this information as an extra paragraph to the Materials and Methods Section on the marine model (currently Sect. 2.2).

Lines 190-195: I suggest removing the part on oxygen. It is no relevant for the study, and it seems to me that per-45 sistent hypoxia/anoxia in the deep basins of the Baltic Sea is mixed up with seasonal hypoxia in parts of the coastal zone.

> removed

Fig. 4: Could be merged with Fig. 3 or at least put as Fig. 3b (ideally with the map inset shown in Fig. 3a/now 50 Fig. 3). There is a horizontal line a few pixels above the horizontal border between "*Belt Sea*" and "*Arkona Basin*", which has the same color as the "*Oresund*". Please double-check that there is no error with the region mask in your analyses. 55

> We would prefer to keep the figures separated so that they are closer to the relevant text passages. The figures are planned as one-column figures in the final document. Hence, they will consume less space in the final document compared to the discussion version. Thanks for looking in details into the plot and identifying the horizontal line. The plotting programm bilinearly interpolated between the masks. We manually draw colored boxes atop of the interpolated regions, which we did not do properly. The issue is fixed now. It only affected this figure and no data.

Lines 208/209: Please state why you picked the three stations that are shown. Do they represent specific regimes? E.g. coastal vs. offshore? Re-evaluate whether you really need the three stations shown in the supplement. If you want to make the validation more sound, then you 70 should point out somewhere in the validation section that model-data agreement is good also at the other stations presented in the supplement. Remove the sentence on the stations that were ignored.

> add text after first sentence: "They represent different 75 regimes in the considered region: two offshore stations in different basins (OMBMPM2 and BY2) and one station close to the shore (DMU547). Additionally, sufficient measurement data were available at these locations.". Replaced the second sentence by "Validation plots at three additional stations are presented in the supplement and show a similar outcome. They are included to indicate that not just the three best stations are shown here.".

**Line 213: Could you specify what bathymetry criteria are used. Perhaps you can add relevant isobaths to Fig. 4?**

85

> added isolines for bathymetry to Fig. 4; the rough spatial extend of the domains/masks was taken from the referenced publication and then fine-tuned based on the used bathymetry

Line 219 and Figs. 5 and 6: I would suggest switching the 90 order of the T and S columns as you first talk about T and then S.

> order of columns switched as suggested

**Lines 222/223: The vertical mixing might be a point for the discussion.**

> Included in paragraph Discussion (first paragraph).

**5 Lines 225-227: These two sentences contradict each other. First "all is good", then nitrate is not.**

> The one sentence ("*all is good*") means the sea surface and the other sentences means the sea floor. Add "...*at the sea floor*" to the first sentence to clarify this.

**10 Lines 239-248: Should this go into the discussion?**

> moved to the discussion

**Lines 244-246: I don't understand this part. Please clarify what is supposed to happen but does not in the model, and how this affects the deposition.**

15 > extended this passage and added reaction equations (now in the Sect. *Summarizing discussion*)

Fig. 8, caption: The last sentence explains how percentiles (10%, 50%, 90%) were calculated for the ratios. However, following this approach, all ratios get the same 20 weight, which may change the results especially during

- periods of strong short-term changes (e.g. during the spring bloom). I would suggest calculating daily (assuming daily model output), spatial integrals of mass (i.e. concentration times volume) of TN and  $TN_{ship}$ . You can
- $_{\rm 25}$  then calculate  $\rm TN_{ship}/\rm TN$  ratios and use the TN mass as weights to calculate weighted percentiles. This way you ensure that short-term changes in concentrations and volume (ERGOM uses a free surface, right?) are accounted for correctly. Analogously, you can calculate the
- 30 weighted percentiles for TN by dividing the daily time series of regions' TN mass by the daily time series of the region' volumes, which gives the spatially weighted-average TN concentration, and using the daily time series of the regions' volumes as weights. Analogously for the other 35 variables.

> For the basin means, we had only monthly mean values available as basis for the calculations. Temporally higher resolved data were only written out at the locations of measurement stations. This was done to save disc space. The plotted
 40 variability rather is a spatial variability than as temporal one.

We added this information to the figure's caption.

**Fig. 10: What is the cause for the strong peaks in the 90% percentiles of PON (and TN) in fall and winter at DMU547?**

> The detritus concentrations are very high at three days in September and December. The high concentrations correlate with peaks in the U and V current velocities at the sea floor. Probably, detritus was resuspended from the sea floor. Values of relevant state variables at different depths at this station are, now, provided in the supplement (dmu547\_peak\_ 50 autumn.pdf);

**Line 292: 6% is not "very high"**

> replaced by "... ratio peaks with  $\approx 6\%$  ... "

Line 297: An introductory sentence why you show the vertically resolved plots would be nice.

55

85

> added two introductory sentences

Lines 303-313: I would suggest rewriting this text part as a "*normal*" paragraph. The bullets make it appear like a list of very important information but they are mostly a detailed description of the dynamics of DIN, DON and PON. If you prefer keeping it as a bullets, please correct the numbering ("3" occurs twice).

> itemization converted into one paragraph

Fig. 11: Would it make sense to add a row for PON since it's referred to a lot on lines 303-313? Perhaps you could 65 label the two transects in Figs. 3/4 (e.g. T1/T2 or S1/S2) and use the label in the figure caption. I was a bit confused about the term *"profiles"*: do you have profile data available? If so, why didn't you use them for validation?

> added T1/T2 to Figs. 3 and 4; added PON to Fig. 11; 70 We have profile data available and we used them to calculate the 10 m-averages in the validation section. However, it was not mentioned explicitly in the *Materials & Methods* section. We added a sentence "*Vertical profiles were measured at most stations*." to the MM Section. 75

Lines 329-332: Please explain the cause of the difference between Oresund and Arkona Basin. Section 3.5: Include references to figures, it's hard to remember what result derives from which figure. Discussion should be in the same order as the results, i.e. annual cycle (lines 329-332) should be discussed before the vertical. And as mentioned in my general comments, please provide an actual discussion.

> added explanation (in the Discussion)

> references to the figures added

> moved paragraph on vertical distribution further downward

| 6 | D. Neumann et al.: Res | ponse to the comments | of Reviewer #1 on | "Contribution of sh | hipping NOx emissions to | " |
|---|------------------------|-----------------------|-------------------|---------------------|--------------------------|---|
|---|------------------------|-----------------------|-------------------|---------------------|--------------------------|---|

| 3 Technical corrections                                                                           |                                                                                                   |    |
|---------------------------------------------------------------------------------------------------|---------------------------------------------------------------------------------------------------|----|
| Line 8: "the atmospheric input"                                                                   | Line 68: "Within ERGOM, the tagging" instead of "Previously, this tagging"                        |    |
| > included                                                                                        | > modified                                                                                        |    |
| Line 9: "in shallow coastal regions"                                                              | Line 71: "and another one without"                                                                |    |
| 5 > replaced                                                                                      | > modified                                                                                        | 35 |
| Line 19: Include reference to MSFD (EU, 2008) for the GES                                         | Line 79: "were not coupled on-line."                                                              |    |
| > included                                                                                        | > modified                                                                                        |    |
| Line 24: remove "approximately"                                                            | Line 98: "deposited"; use full name of SMOKE                                                      |    |
| - > removed                                                                                       | > modified; included                                                                              |    |
|                                                                                                   | Line 99: use full name of STEAM                                                                   | 40 |
| Line 25: "atmospheric nitrogen deposition"                                                        | > included                                                                                        |    |
| > added                                                                                           | Line 100: use full term for IMO                                                                   |    |
| Line 32: "region where high amounts"                                                              | > included                                                                                        |    |
| > modified                                                                                        | Contion of Fig. 1, remove long name of CMAO and the                                               |    |
| 15 Line 43: move HELCOM reference to end of sentence                                              | note                                                                                              | 45 |
| > moved                                                                                           | > removed                                                                                         |    |
| Line 45: "That means"                                                                             | Line 109: Please add in parentheses what HONO is                                                  |    |
| > modified                                                                                        | > HONO is no abbreviation of a substance name like PAN                                            |    |
| Line 46: "ships built after 2021"                                                                 | and PNA but the chemical formula. We reordered the list to have the two abbreviations in the end. | 50 |
| 20 > modified                                                                          | Line 138: remove "past"                                                                           |    |
| Lines 56/57: "in the system and their deposition sites"                                           | > removed                                                                                         |    |
| > modified; but used "the location of their deposition sites" instead of "their deposition sites" | Line 146: "Inorganic" instead of "Basic"?                                                         |    |
| Line 60: "key variables in biogeochemical models"                                                 | > replaced                                                                                        |    |
| 25 > modified                                                                                     | Line 155/156: remove the name of the supplement; it can all be found in the readme.               | 55 |
| Lines 63/64: You also provide analyses for organic N, not just TN and DIN                         | > removed                                                                                         |    |
| • • • • • • • • • • • • • • • • • • •                                                             | Line 157: "by the method"                                                                         |    |
| I manks. We will include it.                                                                      | > modified                                                                                        |    |
| Lines 07/08: remove the sentence "We used an"                                                     | Line 158: "and used"                                                                              | 60 |
| 30 > removed                                                                                      |                                                                                                   |    |

**D. Neumann et al.: Response to the comments of Reviewer #1 on "Contribution of shipping NOx emissions to ..." 7**

> modified

Line 173: "in the eastern"

> modified

Lines 178-180: change order of the two sentences

5 > changed order

Line 181: "emergence of stratification"

> modified

Line 183: remove "read"

> removed

10 Line 185: "N:P" => N and P have not been introduced

> added "nitrogen-to-phosphorus"

Lines 189/190: "algal bloom period ends in autumn when stratification"

> modified

15 Line 193/194: remove "- denoted as oxygen minimum zones -"

> removed

Lines 205/206: *"were aggregated into a monthly 'climatology"* ('climatology' with apostrophes as seven years do 20 not really warrant a proper climatology)

> added

 Table 1, caption: remove "and model evaluation"; remove color references to figures

> removed; reformulated caption slightly

25 Table 1 should come before Fig. 4 as it is referred to first.

> Table 1 is before Fig. 4 in the LaTeX document. We will take care that they will be properly ordered in the final version.

Fig. 5: remove "var [unit] (see header)" on the left; Why 30 are some individual ticks on the y axes not labelled (e.g. 0 in bottom left panel or 15 in the one next to it)? Use "mmol N m-3" and "mmol P m-3" as units for NO3 and PO4, respectively; caption: remove color references; "Each column presents one state variable"; "Vertical lines

**and shaded areas show the monthly"; remove everything 35 after the first "Supplement"**

> removed "var [unit] (see header)"; Default plotting in R – better this way (adjusted minimum of y-axis; decreased font size of two plots; manually added some values along the y-axis)? Otherwise the labels would have a too small font 40 size; We would like to keep the units SI conformal but 'N' and 'P' are not defined by SI. As long was we have amount ('moles') and not mass ('g'), *x* mol of N are equal to *x* mol of nitrate; colored references removed; column/row corrected; "*the*" in the sentence on "*vertical lines and shaded areas*" 45 removed; removed text after "*Supplement*";

**Line 212: "Basin definitions by Omstedt"**

> modified

Line 215: "the model's open boundary"

> modified

50

Line 219: "Sea surface temperature [...] but sea surface salinity"

> removed "*the*"

Fig. 6: remove "var [unit] (see header)" on the left; Why are some individual ticks on the y axes not labelled (e.g. 55 15 in bottom left panel)? Use "mmol N m-3" and "mmol P m-3" as units for NO3 and PO4, respectively; Why are the depth ranges at the first 2 stations not equivalent to 10m as stated on line 198? Caption: "Same as Fig. 5 but for the bottom 10 m."

> adapted like in Fig.5; In shallow regions below  $\approx 30~{\rm m}$  less than 10 m (above the bottom) were considered. Statement in 1. 198 was correct; first sentence simplified similar as suggested

Line 222: Here and throughout the manuscript, do you 65 mean "seasonal" with "intra-annual"? Since you show monthly data, any shorter-term variability is averaged out.

> Yes, we meant "*seasonal*". We replaced "*intra-annual*" by "*seasonal*".

Lines 227/228: replace "The modeled water column is stronger stratified than the real water column" with "Simulated salinity suggests that stratification is overestimated by the model"

> replaced

70

Fig. 7: Should the unit be "*mmol m-3*", not "*µmol m-3*"? Remove the "*all*" subscript; stations/transects hard to see in grey-scale; caption: remove color references; "*White areas are on land in the MOM-ERGOM domain.*"

5 > Yes, should be "*mmol m-3*"; We think that it is a good idea to add a subscript to all TN to be unambiguous; removed color references;

Lines 239-241: "origin and caused by a lower horizontal grid resolution of the CMAQcompared to MOM-ERGOM 10 and interpolation over the land-sea interface."; remove the last sentence

> modified

Line 247: remove "to the ground/sea"

> removed

15 Line 249: here and later: *"The contribution of shipping-related nitrogen to TN* (TNship/*TN*)..."

> resolved here but not at later locations

Fig. 8: I would suggest removing the absolute shipping TN, DIN, etc. since it is pretty much zero compared to the 20 overall quantities and hardly visible especially in grey-scale. The y axes labels could then be "*overall*" for the even rows and "*shipping*"; I suggest putting TN in the first column as you previously analyzed TN, so now you increase the level of detail from TN to its inorganic and 25 organic compartments; add to the caption that it's 2012 only and remove "*in 2012*" from the x axes labels; caption: "*thick*" instead of "*think*"; remove "*in the odd rows*"; "*For the ratios*, …"; remove color references

> We would like to keep the green lines in odd rows be 30 cause they make hugh difference between untagged nitrogen and shipping-related nitrogen clear and do not disturb the understanding of the plots.

> order of DIN, DON, PON, and TN: DIN is described first and TN last in Sect. 3.3. Hence, it is reasonable to keep
 35 the order as it is. For the same reason, the *temperature* and *salinity* were switched in Figs. 5 and 6.

> replaced "think" by "thick"

> kept color reference but added description in terms of darker/lighter color

**40 Line 255: remove "with all nitrogen" and "all" subscript**

> removed "*with all nitrogen*" but kept the subscript to prevent ambiguity

Lines 259/260: "which are the sum of DIN, DON and PON"

> modified

Line 261: "In contrast, the DIN concentrations are elevated ( $\approx 5 \text{ mmol N m}^{-3}$ ) throughout the year in the Oresund ."

> modified

Lines 262/263: "*by riverine nutrient loads*"; remove the 50 names of the rivers listed in parentheses

> removed

Lines 264/265: "The relative contributions of shipping N to DIN, DON and PON are very small."

> modified

55

75

45

Lines 267/268: "from 1.5-2% in January to about 1% in July"

> modified

Fig. 9: These are no daily values (see caption); put only one station/depth label per station (like in Fig. 8: change caption to: "Same as Fig. 8 but for specific stations (see Fig. 1)."

> The sentence was formulated ambiguously. We have daily mean values and calculate monthly percentiles from them. In Fig. 8, we had only monthly mean values. Background: We saved model output at specific locations (measurement stations) in daily resolution. But, the full spatial model output was only stored in monthly resolution to space disc space. Fig. 9 shows the variability in time, whereas Fig.8 shows variability in space.

**Line 274: "However" instead of "But"**

> modified

Line 275: remove "as presented in Sect. 3.2"; "in the open ocean"

> removed and modified

**Line 278: These are no means and no daily data**

> adapted to be less ambiguous; see reply to comment on Fig. 9; Fig. 10: put only one station/depth label per station (like in Fig. 8); again why I sthe depth range not equal to 10 m for the first two stations? Change caption to: "Same as Fig. 9 but for the bottom 10 m."

5 > station label: updated; depth range: see reply to comment on Fig. 6; modified caption similar as suggested;

Line 286: remove "data"

> removed

Line 291: "at the surface due to vertical stratification."

> modified 10

Line 297: State what quantities are shown.

> added; also included information on temporal resolution

Line 299: "causing the low values in the south."

> modified

Line 300: "later" instead of "delayed"

> modified

Line 303: remove "reaches  $\approx 12.5\%$ "

> removed

20 Fig. 11: add "latitude (N)" to x axis on left and right; station lines are barely visible in grey scale, same for the temporal development; caption: remove "is plotted", remove last sentence

> added 'latitude (N)'; changed color of station-symbol 25 and -line; improvement of color-scale does not seem to be possible when it should remain equal for all 24 plots; removed both:

**Line 315: Please rewrite the sentence such that models and data are only mentioned once**

> rewritten 30

Line 317: "The concentration of shipping-related TN..."

> modified

Line 320: "... the contribution of shipping-related N to DIN was highest ...."

> modified 35

Line 334: "to TN" > modified Line 335: "the NOx emissions" > modified Line 337: "to DIN" > modified Line 349: "." after ")" > added Line 368: add first name of co-author (although I would suggest to only use initials in the whole author contribu- 45 tion section)

40

> changed to: only initials

Line 383: "program whose intense"

> modified

Daniel Neumann1, Matthias Karl2, Hagen Radtke1, Volker Matthias2, René Friedland3, and Thomas Neumann1

1Leibniz-Institute for Baltic Sea Research Warnemünde, Seestr. 15, 18119 Rostock, Germany
 2Institute of Coastal Research, Helmholtz-Zentrum Geesthacht, Max-Planck-Str. 1, 21502 Geesthacht, Germany
 3European Commission DG Joint Research Centre, Directorate D – Sustainable Resources, Via Fermi, 2749 – TP 270, I-21027 Ispra (VA), Italy

Correspondence: D. Neumann (daniel.neumann@io-warnemuende.de.de)

**Response to the comments of Reviewer #2**

We thank the reviewer for the constructive comments on the manuscript. The reviewer's comments are written in **bold font**. The authors' replies start with a ">" and are written 5 in normal font.

**1 General comments**

The manuscript attempts to answer a relevant ocean research question, which is very clearly stated in the title.

The manuscript reads in a clear, concise, and well-10 structured way. The scientific approach is transparent and the methods and results are presented in an appropriate way.

However, the manuscript is lacking in the discussion and conclusions sections.

> see below our reply to **Specific comments**

**2 Specific comments**

The summarizing discussion section mainly consists of a summary of the results and very little actual discussion and the results are not set in context to relative literature.

20 The conclusions section partly consists of discussion and recommendations for further studies. It is not very clear

what the conclusions are, except "..., the shipping sector might relevantly contribute to eutrophication at specific locations in the wester Baltic Sea in summer." It seems too unsubstantial for the work that has been 25 done and needs to be improved.

> A new "*Discussion*" section has been added, in which we discuss relevant aspects. The "*Summarizing Discussion*" section has been integrated into this new section and has been modified. Some paragraphs of the conslusions have been replaced by new ones.

**3** Technical corrections**

Page 2, Line 33: I think it should be 'where' instead of 'with'.

| > | rep | laced | as | suggested |  |
|---|-----|-------|----|-----------|--|
|---|-----|-------|----|-----------|--|

35

40

Page 3, Line 79: Remove 'But' at the start of the sentence.

> removed as suggested

Page 16, Line 288: Consider rephrasing "... stations distant to the coast ...".

> repleed by "stations in the center of the basins"

[revised manuscript text omitted]
 contrastin the Öresund, the DINall concentration remain at elevated concentrations concentrations are elevated ( $\approx 5 \text{ mmol m}^{-3}$ ) throughout the year . The Öresundis considerably impacted by nutrient loads from the Swedish mainland (Braån, Saxån, Kävlingeån, and Höje å Rivers) and from Zealand (Mølleåen River). This might cause the differing intra-annual pattern. The intra-annual in the Öresund. The seasonal patterns of the DONall and PONall concentrations are the same as at the other stations. The relative contributions of shipping N to DIN, DON and PON are very small.
- 320 The absolute , , and concentrations are very low compared to the , , and concentrations.

In the Belt Sea, the intra-annual seasonal variability of the shipping contribution and its spatial variability are very low in all nitrogen fractions. The shipping contribution is between 1% and 2%. In the Öresund, it decreases from approximately 1.5% to 2% in January towards approximately to about 1% in July and then increases again towards the end of the year. Finally in the Arkona Basin, the shipping contribution increases from the beginning of the year until summer and then decreases. The values

325 are in a range between 1% and 4.5%. However, there are some places in the Arkona Basin where the shipping contribution remains below 2%.

Summarizing, the three considered basins represent three different regimes of shipping-related nitrogen deposition and of its contribution to the biogeochemical cycle. ButHowever, the relevance of shipping-related nitrogen differs spatially within each basinas presented in Sect. 3.2: the shipping contribution to the nitrogen fractions is much higher at in the open ocean than along

330 the coastline. Hence, the three stations from the validation, two of which are open ocean stations, are taken again to assess the intra-annual variation of the shipping contribution at open-sea locations.

Figures 9 and 10 show monthly median and percentiles of calculated from daily mean values at three stations the three stations (two of which are in the open ocean) in the surface and bottom layer, respectively. At the sea surface, the intra-annual seasonal cycles of the DINall, DONall, and PONall concentrations are as expected. The time series of PONall and TNall concentrations shows two peaks: the first is the diatom bloom in spring and the second a cyanobacteria bloom in later summer.

335

340

Cyanobacteria do not grow in the northern Belt Sea and Kattegat because the salinity is too high.

In the surface layer, the relative shipping contribution rises in all fractions and at all stations in spring, peaks in summer, and decreases again. At BY2, the  $PON_{ship}/PON_{all}$  ratio decreases already after June and has a minimum in August, after which it increases again. The minimum is caused by the cyanobacteria bloom because the cyanobacteria fixate non-tagged N2. The overall shipping contribution at DMU547 is similarly low as in the total basindate. At OMBMPM2 and BY2, the  $TN_{ship}/TN_{all}$

ratio exceeds 5%. At BY2, the  $DIN_{ship}/DIN_{all}$  ratio even exceeds 10%. Thus, the shipping-related nitrogen contribution in summer is much higher at individual stations distant to the coast in the center of the basins than on basin average – in the surface layer.

The shipping contribution to the nitrogen fractions is much lower in the bottom layer of the three stations. It remains below 2% in all nitrogen fractions at DMU547 and OMBMPM2. At BY2, the contribution is higher than 2% but still considerably lower than at the surface . The vertical stratificationmentioned in the validation section (Sect. 3.1) causes thisdue to vertical stratification. The PONship/PONall ratio becomes very high 
[revised manuscript text omitted]

---

## Editor Decision (ED1)

Line 82: "Nitrogen" twice at the beginning of the sentence. Suggest "Shipping related nitrogen deposition…"

Lines 145-156: "phytoplankton" is used as a plural noun except when referring to a single organism; the verbs therefore need changing to plural in several places

Line 151: change word order "The growth of cyanobacteria depends only on…"

Line 196: Insert "there": "In 2012 there were no…"

Line 246: "…measurements show a decrease…" (not "shows")

Line 257: "emissions" not "immissions"

Line 287: "fix" not "fixate"

Line 335: "ratio" not "fraction"

Line 350: "floor" not "flor"

Line 381 (equation R1): the subscript (aq) should be placed at the end of the formula $H_3O^+$

Line 386: "sodium" not "natrium"

Line 397: "..distant from the coast.." not "..distant to the coast.."

Lines 397-398: sentences needs to be clearer, e.g. "This agreement results not from shipping activity but rather from the lack of riverine nitrogen sources in offshore regions."

Line 438: typo "subtracted"

Line 442: "valid" occurs twice here: best to delete the first occurrence of "valid"

Line 444: the text is written in US English, so change "behaviour" to "behavior"

Line 453: typo "absolute"

Figure 7: The reponse to the Reviewer stated that the cross sections are marked as in Figures 3 and 4, i.e. with the Arkona cross section should be shown as a dashed line